# Initializing Services in Interactive ML Systems for Diverse Users

**Avinandan Bose**
University of Washington
avibose@cs.washington.edu

**Mihaela Curmei**
University of California Berkeley
mcurmei@berkeley.edu

**Daniel L. Jiang**
University of Washington
danji@cs.washington.edu

**Jamie Morgenstern**
University of Washington
jamiemmt@cs.washington.edu

**Sarah Dean**
Cornell University
sdean@cornell.edu

**Lillian J. Ratliff**
University of Washington
ratliffl@uw.edu

**Maryam Fazel**
University of Washington
mfazel@uw.edu

## Abstract

This paper investigates ML systems serving a group of users, with multiple models/services, each aimed at specializing to a sub-group of users. We consider settings where upon deploying a set of services, users choose the one minimizing their personal losses and the learner iteratively learns by interacting with diverse users. Prior research shows that the outcomes of learning dynamics, which comprise both the services' adjustments and users' service selections, hinge significantly on the initial conditions. However, finding good initial conditions faces two main challenges: (i) *Bandit feedback:* Typically, data on user preferences are not available before deploying services and observing user behavior; (ii) *Suboptimal local solutions:* The total loss landscape (i.e., the sum of loss functions across all users and services) is not convex and gradient-based algorithms can get stuck in poor local minima.

We address these challenges with a randomized algorithm to adaptively select a minimal set of users for data collection in order to initialize a set of services. Under mild assumptions on the loss functions, we prove that our initialization leads to a total loss within a factor of the *globally optimal total loss with complete user preference data*, and this factor scales logarithmically in the number of services. This result is a generalization of the well-known $k$-means++ guarantee to a broad problem class, which is also of independent interest. The theory is complemented by experiments on real as well as semi-synthetic datasets.

## 1   Introduction

We consider a setting where a provider wants to design $k$ services for $n$ users with diverse preferences. Each service uses a model parameterized by a vector $\theta \in \mathbb{R}^d$ to predict users' preferences, and users pick a service that yields the smallest loss for them. The loss incurred by user $i$ when choosing a service parameterized by $\theta$ is denoted by $\mathcal{L}_i(\theta, \phi_i)$, where $\phi_i \in \mathbb{R}^d$ parameterizes the user's preference. We want to design $k$ services by minimizing the sum of all losses; i.e., an optimization

38th Conference on Neural Information Processing Systems (NeurIPS 2024).

problem of the form:

$$\underset{\theta_1,\ldots,\theta_k \in \mathbb{R}^d}{\text{minimize}} \sum_{i=1}^{n} \min\{\mathcal{L}_i(\theta_1, \phi_i), \ldots \mathcal{L}_i(\theta_k, \phi_i)\}. \tag{1}$$

This problem formulation is broad and includes the classical $k$-means clustering problem [28] (where $\mathcal{L}_i$ are Euclidean distances and the inner 'min' selects the closest among $k$ centroids), mixed-linear regression [42], generalized principal component analysis (GPCA) or subspace clustering [1], in addition to our new, motivating problem of designing $k$ services for $n$ users. Even if the losses $\mathcal{L}_i$ are convex in $\theta$, this objective is generally not convex (even in the special case of the $k$-means problem).

Our goal is to find a local minimum of this optimization problem with an approximation ratio (a worst-case guarantee on the achieved total loss with respect to the global optimum) under suitable yet broad assumptions on $\mathcal{L}_i$. Further, an important limitation in many practical settings is that the provider/designer has only *bandit feedback* (zeroth-order oracle access) to the loss functions (i.e., the designer doesn't know the function $\mathcal{L}_i(\cdot, \phi_i)$, but can only evaluate its value for some $\theta$ corresponding to the service chosen by the user among the ones deployed), which further complicates solution methods, compared to the classical cases of clustering and facility location problems [10] which typically assume full information.

In this paper, we seek a novel and effective initialization scheme that vastly extends the celebrated $k$-means++ algorithm and its analysis [5]. This scheme should retain the simplicity and ease of implementation of the original algorithm, yet be able to (1) handle general loss families (assumptions on $\mathcal{L}_i$ are discussed in Section 2), (2) provide a tight, instance-dependent approximation ratio (details in Section 3), and (3) handle realistic information limitations such as access only to (noisy) bandit feedback, in a sample-efficient manner. Next, we describe in more detail the important use case of initialization of services for diverse users in multi-service ML systems (yet as noted above, our main result has other applications as well, and can be of independent interest).

**Motivation.** In a variety of contexts such as federated learning [27], crowd-sourcing [36] and online recommendation systems [35], data about user preferences is acquired through iterative interactions. This data is then used to improve the model and serve the individual needs of users. Given that users' preferences are typically heterogeneous, recent works demonstrate that using multiple specialized models can be more effective than the one-size-fits-all approach of employing a single large shared model, e.g., for clustered federated learning [30, 34, 17], meta learning [25, 7], fine-tuning for specific groups of users or tasks [9, 37], and in the context of fair classifiers [39]. Here we tackle the crucial yet under-explored phase of *initializing services* in ML systems that learn interactively from diverse users. The initialization process is crucial as it sets the stage for how effectively these systems can adapt and specialize with future user interactions. Once initialized, the services interact with users, who, in turn, choose among services based on their loss. These "learning dynamics" typically lead to the specialization of services to groups of users [18, 12], and [12] shows experimentally that the overall social welfare achieved by the services depends on the initialization of the learning dynamics. We note that in the context of problem (1), the learning dynamics in [12] can be seen as updates of an alternating minimization algorithm, iteratively updating users' choices (the inner minimization) and services' parameters (update to each $\theta_j$). Our goal is to initialize a set of services to minimize the sum of losses for all users (or equivalently, maximize total welfare), tackling the following challenges:

- **Bandit loss feedback:** In practice, offering a service often precedes data collection. Specifically, in contexts like online recommendations, it is usually not feasible to gather user preference data (knowledge about $\phi_i$ and evaluations of $\mathcal{L}_i(\cdot, \phi_i)$ at various different parameter values in problem (1)) without first deploying the services parameterized by $\{\theta_1, \ldots, \theta_k\}$ and observing user interactions. This means that data collection is inherently conditional on the existence of services, challenging the conventional "data-first, model-second" paradigm.

- **Suboptimal local solutions:** Since users select the service with the lowest loss $(\min\{\mathcal{L}_i(\theta_1, \phi_i), \ldots \mathcal{L}_i(\theta_k, \phi_i)\}$ in problem (1)), minimizing over the parameters $\{\theta_1, \ldots, \theta_k\}$ leads to a nonconvex problem in general, as mentioned earlier. Gradient-based learning dynamics can get stuck in local minima where the total loss can be significantly worse than the globally minimum loss. Thus, the outcomes of learning dynamics are heavily affected by initialization of service parameters.

**Contributions.** The following summarizes the contributions of this paper.

- We design a computationally and statistically efficient algorithm for *initialization of services* prior to learning dynamics. The algorithm works by adaptively selecting a small number of users to collect data from (via queries of their loss function) in order to initialize the set of services.

- We establish an *approximation ratio* for the designed algorithm: the expected total loss achieved by the algorithm right after initialization is within a factor of the *globally optimal total loss in the presence of complete user preference data*, and this factor scales logarithmically in the number of services. Furthermore this bound is tight, and recovers the known k-means++ approximation ratio as a special case (cf. Section 3).

- When users belong to a set of demographic groups, it is desirable that the services do not result in unfavorable outcomes towards certain demographics (e.g., based on gender or racial groups). One fair objective is to minimize the maximum average loss of users across different groups. We provide an *approximation ratio* for this fair objective that scales logarithmically in the number of services (cf. Section 3).

- In the context of linear prediction models, we study the problem of *generalizing* to users the provider has not interacted with before (cf. Section 4).

- We empirically demonstrate the strengths of our initialization scheme via experiments on a prediction task using 2021 US Census data, and online movie recommendation task using the Movielens10M dataset (cf. Section 5).

## 1.1 Related Work

**Multiple Model Specialization.** In distributed learning, where data sources are users' personal devices, utilizing multiple specialized models, where users are grouped into clusters representing interests, can yield improved predictions and outcomes. For instance, in recommendation systems these clusters could represent users interested in different movie genres, or different combination of features (see Appendix B for a concrete example on Netflix recommendation clusters). This approach has been adopted recently in clustered federated learning [34, 30, 17] and online interactive learning [32], facility location problems [6], where users *choose* models/services and for which they provide updates.

**Clustering.** Multiple model specialization leads to clustering the users into groups and centering a specialized model on each group. We provide a brief review of the $k$-means clustering problem and establish the connection to specialization. The $k$-means clustering problem is one of the most commonly encountered unsupervised learning problems. Given a set of known $n$ points in Euclidean space, the goal is to partition them into $k$ clusters (each characterized by a center), such that the sum of square of distances to their closest center is minimized. Dasgupta [11] and Aloise et al. [3] showed that the $k$-means problem is NP-Hard. The most popular heuristic for $k$-means is Lloyd's algorithm [28], which proceeds by randomly initializing $k$ centers and then uses iterative updates to find a locally optimal $k$-means clustering, which can be arbitrarily bad compared to the globally optimal clustering. The performance of the $k$-means algorithm relies crucially on the initialization. Arthur and Vassilvitskii [5] and Ostrovsky et al. [33] proposed an elegant polynomial time algorithm for initializing centers, known as $k$-means++. Arthur and Vassilvitskii [5] proved that the expected cost of the initial clustering obtained by $k$-means++ is at most $8(2 + \log k)$ times the cost of optimal $k$-means clustering. Our work generalizes the analysis of Arthur and Vassilvitskii [5] to the setting where *user's preferences are represented as unknown points and the loss functions are unknown with only bandit access, not necessarily identical, and general as long they satisfy Assumptions 2.1 and 2.2*, with important examples given in Appendix C.

For a detailed discussion on more related works please see Appendix A.

**Notation and Terminology.** For a symmetric matrix $\mathbf{A}$ and any vector $x \in \mathbb{R}^d$, we denote its Mahalanobis norm by $\|x\|_{\mathbf{A}} = \sqrt{x^\top \mathbf{A} x}$. The generalized eigenvalues for a pair of symmetric matrices $\mathbf{A}$ and $\mathbf{B}$ are denoted by $\lambda(\mathbf{A}, \mathbf{B})$, defined as the solutions of $\lambda$ for the generalized eigenvalue problems $\mathbf{A}v = \lambda \mathbf{B}v$ [16, 14]. Specifically we use $\lambda_{\min}(\mathbf{A}, \mathbf{B})$ to denote minimum generalized eigenvalue for the matrix pair $\mathbf{A}, \mathbf{B}$. The loss for a user $i$ given service $\theta \in \mathbb{R}^d$ is denoted by $\mathcal{L}_i(\theta, \phi_i)$ where $\phi_i$ parameterizes the user's preference. For a set of users $\mathcal{A}$ (e.g., $\mathcal{A} = [n]$ denotes a set of $n$ users) and a set of services $\Theta = \{\theta_1, \ldots, \theta_k\} \subset \mathbb{R}^d$, the total loss is defined as $\mathcal{L}(\Theta, \mathcal{A}) = \sum_{i \in \mathcal{A}} \min_{j \in [k]} \mathcal{L}_i(\theta_j, \phi_i)$.

## 2 Problem Setup

We make the following assumptions about the functional form of $\mathcal{L}_i$, and state several examples of function classes satisfying these properties. Note that the designer/provider doesn't need to know the functional form of $\mathcal{L}_i$, and knowledge about $\mathcal{L}_i$ is obtained through bandit feedback via observing the scalar values $\mathcal{L}_i(\theta, \phi_i)$ for different $\theta$, where $\theta$ parameterizes the services.

**Assumption 2.1** (Unique Minimizer). The loss function satisfies the following equivalence: $\mathcal{L}_i(\theta, \phi_i) = 0 \iff \theta = \phi_i$.

This assumption implies that unless all users have identical preference parameters, there doesn't exist a single service parameter $\theta$ that simultaneously minimizes every user's loss. Thus providing multiple services (multiple $\theta$'s) where the users choose the one best for them is strictly better than one service for all users.

**Assumption 2.2** (Approximate Triangle Inequalities). For a pair of users $i, j$ there exists a finite constant $c_{ij} > 0$ such that for all $\theta \in \mathbb{R}^d$ the following hold:

    *(i)* $c_{ij}\mathcal{L}_i(\theta, \phi_i) \leq \mathcal{L}_j(\theta, \phi_j) + \mathcal{L}_j(\phi_i, \phi_j)$.

    *(ii)* $c_{ij}\mathcal{L}_i(\phi_j, \phi_i) \leq \mathcal{L}_j(\theta, \phi_j) + \mathcal{L}_i(\theta, \phi_i)$.

Here $c_{ij}$ (equal to $c_{ji}$) captures the alignment between the preference parameters and loss geometries of two users. Lower values of $c_{ij}$ indicate less similarity. Item $(i)$ implies that the loss for user $i$ on any service $\theta \in \mathbb{R}^d$ is no worse than (up to a constant factor) the sum of (a) loss of another user $j$ on using the same service, and (b) the loss of user $j$ if they were to use user $i$'s preference parameter. The latter term (b) can be seen as measuring the similarity between the users' preferences.

For condition $(ii)$ to hold for all $\theta \in \mathbb{R}^d$, it must also hold for the service that minimizes the sum of losses of both users using the same service. Suppose that users $i$ and $j$ were to exchange preference parameters. Then their loss would be no worse than (up to a constant factor) their minimum total loss, i.e.,

$$c_{ij}\mathcal{L}_i(\phi_j, \phi_i) \leq \min_{\theta \in \mathbb{R}^d} \left( \mathcal{L}_j(\theta, \phi_j) + \mathcal{L}_i(\theta, \phi_i) \right).$$

Some examples of loss functions and the corresponding constants that satisfy these assumptions include the following (see Appendix C for additional examples and derivations):

- Squared error loss for linear predictors (cf. Section 4).
- The Huber loss on the prediction error:

$$\mathcal{L}_i(\theta, \phi_i) = \begin{cases} \frac{1}{2}\|\theta - \phi_i\|^2, & \text{if } \|\theta - \phi_i\| \leq \delta \\ \delta(\|\theta - \phi_i\| - \frac{1}{2}\delta), & \text{otherwise.} \end{cases}$$

    This loss is used typically in robust estimation tasks. Here $\|\cdot\|$ could be any norm, and we show $c_{ij} = 1/3$.
- The normalized cosine distance: $\mathcal{L}_i(\theta, \phi_i) = 1 - \theta^\top \phi_i$ where $\|\theta\|_2 = \|\phi_i\|_2 = 1$, with $c_{ij} = \frac{1}{2}$. This is commonly used as a similarity measure in natural language processing applications, for example finding similarity between two documents.
- The Mahalanobis distance: $\mathcal{L}_i(\theta, \phi_i) = \|\theta - \phi_i\|_{\Sigma_i}$. Different users can have different $\Sigma_i$ capturing their diverse loss variation, as long as $\Sigma_i$ is full rank. Here $c_{ij} = \min\{\lambda_{\min}(\Sigma_i, \Sigma_j), \lambda_{\min}(\Sigma_j, \Sigma_i)\}$.
- Any distance metric: This naturally follows from triangle inequality, hence $c_{ij} = 1$.
- Any arbitrary function $\mathcal{L}_i(\theta, \phi_i)$ that is $L_i$-Lipschitz and $\mu_i$-strongly convex in $\theta$ with $c_{ij} = \min(\mu_i, \mu_j)/\max(L_i, L_j)$.

**Objective.** Suppose the users have access to $k$ services parameterized by $\Theta = \{\theta_1, \ldots, \theta_k\} \subset \mathbb{R}^d$. Then, each user $i$ selects service $\theta_l$ that minimizes their loss, i.e. $\mathcal{L}_i(\Theta, \phi_i) = \min_{l \in [k]} \mathcal{L}_i(\theta_l, \phi_i)$.

As discussed earlier, our goal is to design $\Theta$, such that the sum of losses across users and services is minimized. We define the objective as follows:

$$\mathcal{L}(\Theta, [n]) = \sum_{i \in [n]} \min_{j \in [k]} \mathcal{L}_i(\theta_j, \phi_i) = \sum_{i \in [n]} \mathcal{L}_i(\Theta, \phi_i). \tag{2}$$

**Definition 2.3.** Define the unknown optimal set of $k$ services that minimizes the objective to be

$$\Theta_{\text{OPT}} := \text{argmin}_{|\Theta|=k} \mathcal{L}(\Theta, [n]).$$

---

**Algorithm 1** AcQUIre- Adaptively Querying Users for Initialization

---

1: **Input:** Set of users $[n]$, number of services $k$.
2: Choose a user $i$ uniformly randomly from $[n]$.
3: Query user $i$'s preference $\phi_i$, set the first service $\Theta_1 = \phi_i$.
4: **for** $t \in \{2, \ldots, k\}$ **do**
5:     **User behavior:** Collect user losses on existing services $\Theta_{t-1} : \{\mathcal{L}_i(\Theta_{t-1}, \phi_i)\}_{i \in [n]}$.
6:     **User Selection:** Sample $l$ from $[n]$ with probability $P(l = i) \propto \mathcal{L}_i(\Theta_{t-1}, \phi_i)$.
7:     **New service:** Query user $l$'s preference $\phi_l$.
8:     $\Theta_t = \Theta_{t-1} \cup \phi_l$.
9: **end for**
10: **Return** $\Theta_k$

---

Specifically, $\Theta_{\mathrm{OPT}}$ defines a "clustering", meaning a partitioning of the $n$ users into $k$ clusters. The cluster $\mathcal{B}_m$ is the set of all users that prefer the service $\theta_m$ among all the services in the optimal set $\Theta_{\mathrm{OPT}}$. In other words, $\mathcal{B}_m$ is defined as the set of all points such that $\mathcal{B}_m = \{i \in [n] \mid \theta_m = \arg\min_{\theta_l \in \Theta_{\mathrm{OPT}}} \mathcal{L}_i(\theta_l, \phi_i)\}$. If multiple services are equally preferred by a subpopulation, the ties are broken arbitrarily. The resulting set of clusters is denoted by $\mathcal{C}(\Theta_{\mathrm{OPT}}) = \{\mathcal{B}_1, \ldots, \mathcal{B}_k\}$.

The are several statistical and computational challenges to this problem.

*Challenge* 1. Since preferences $\{\phi_i\}_{i \in [n]}$ and loss functions $\{\mathcal{L}_i\}_{i \in [n]}$ are unknown and the provider only has zeroth order or bandit feedback access, estimating the objective function $\mathcal{L}(\Theta, [n])$ usually needs a lot of data collected uniformly across the users. This large amount of data is needed before services can be deployed, yet as stated earlier, we are in the situation where we have no data until we deploy services and observe user interactions. Our limited access to user information (via limited queries) makes our setting challenging.

*Challenge* 2. The loss function is non-convex and iterative minimization approaches from a random initialization are susceptible to getting stuck in arbitrarily poor local optima. This means computing the optimal clustering first and then finding the best service for each cluster is NP-Hard.

*Challenge* 3. We do *not* assume any data separability conditions, for example user preference parameters are drawn from $k$ well separated distributions. Thus we are unable to exploit underlying structure to reduce sample complexity.

Despite the challenges, in Section 3, we propose an algorithm that is both statistically and computationally efficient, and admits an approximation ratio with respect to the *globally optimal value*, i.e., $\mathcal{L}(\Theta_{\mathrm{OPT}}, [n])$.

## 3 Algorithm & Main Results

In this section, we present our initialization algorithm with guarantees, Algorithm 1, and describe how the steps of the algorithm arise naturally in the interactive systems under consideration. Since collecting data uniformly across all the $n$ users can be prohibitively expensive, our goal is to get data from a minimal number of users.

Each iteration of the loop in the algorithm adds a service sequentially and the loop terminates when there are $k$ services, where $k$ is a predetermined parameter for the algorithm. We focus on the loop (lines 4-8) in Algorithm 1.

Suppose at time $t - 1$, the set $\Theta_{t-1}$ is the set of current $t - 1$ services. Then, at time step $t$ the following steps take place.

- **User behavior (line 5).** Given the list of services $\Theta_{t-1}$ users are assumed to choose the best service that minimizes user loss. Users report their losses with respect to the service they choose from the set of existing services $\{\mathcal{L}_i(\Theta_{t-1}, \phi_i)\}_{i \in [n]}$. In practice, this step requires deploying the services and collecting signals of engagement and utility to determine the loss associated with each user, under the behavioral assumption that users are rational agents that choose the best available service. The provider thus needs to measure each user's loss **only** in their single chosen service.
- **User selection (line 6)** A new user $l$ is selected with probability proportional to $\mathcal{L}_i(\Theta_{t-1}, \phi_l)$. This ensures that users that are currently poorly served by existing services are more likely to be selected.

- **New Service (line 7-8).** Given a selected user $l$, the algorithm queries the preference $\phi_l$ of the user and centers the new service at that preference $\theta_t = \phi_l$. In practice, this step requires acquiring data about the user in order to learn their preference parameter (this is needed for only $k$ total users throughout the algorithm). For example, data may be acquired by incentivizing the selected users, via offering discount coupons or free premium subscriptions [21].

With each iteration the loss of each user is non-increasing; the previous services remained fixed and a user would switch to a new service only if it improves quality or equivalently decreases loss. Since at each iteration, a new service is added, the process terminates after $k$ steps. Since it is costly to offer and maintain too many different services, we typically have $k \ll n$.

We now discuss the theoretical properties of the set of services we get at the termination of Algorithm 1.

**Theorem 3.1.** *Consider $n$ users with unknown preferences $\{\phi_1, \ldots, \phi_n\} \subset \mathbb{R}^d$, and associated loss functions $\mathcal{L}_i(\cdot, \cdot)$ satisfying Assumptions 2.1 and 2.2, with bandit access. Let $\Theta_{\mathrm{OPT}} \subset \mathbb{R}^d$ be the set of $k$ services minimizing the total loss and $\mathcal{C}(\Theta_{\mathrm{OPT}})$ the resulting partitioning of users (Definition 2.3). If Algorithm 1 is used to obtain $k$ services $\Theta_k$, then the following bound holds:*

$$\mathbb{E}_{\Theta_k}[\mathcal{L}(\Theta_k, [n])] \leq K_{\mathrm{OPT}}(2 + \log k) \cdot \mathcal{L}(\Theta_{\mathrm{OPT}}, [n]),$$

*where the expectation is taken over the randomization of the algorithm and $K_{\mathrm{OPT}}$ is equal to*

$$\max_{\mathcal{B} \in \mathcal{C}(\Theta_{\mathrm{OPT}})} \frac{4}{\min_{j \in \mathcal{B}} \sum_{i \in \mathcal{B}} c_{ij}} \left( \max_{j \in \mathcal{B}} \sum_{i \in \mathcal{B}} \frac{1}{c_{ij}} \right). \tag{3}$$

A detailed proof is presented in Appendix D; we summarize the main ideas here. The intuition is that a chosen user's preference parameter is typically a good representative for other users in its cluster. Thus adding a service parameterized by the chosen user's preferences generally reduces the losses of users in this cluster. Subsequently we are less likely to pick another user from the same cluster. The $\log k$ factor is due to clusters from which users were never picked.

A similar proof approach was used by Arthur and Vassilvitskii [5] in the context of the $k$-means problem, by sequentially placing centers on *known* points sampled with probability proportional to the point's squared distance to its closest existing center. A key novelty of our analysis is to capture the *alignment* of diverse loss geometries across users in a large class of functions, specifically understanding how user similarities $c_{ij}$ affect the approximation ratio.[1]

**Key characteristics of $K_{\mathrm{OPT}}$:** The following are essential characteristics of the term $K_{\mathrm{OPT}}$.

(i) All terms in $K_{\mathrm{OPT}}$ depend on the local clusters in the unknown optimal clustering $\mathcal{C}(\Theta_{\mathrm{OPT}})$.

(ii) The constant $\min_{j \in \mathcal{B}} \frac{1}{|\mathcal{B}|} \sum_{i \in \mathcal{B}} c_{ij}$ captures the user whose loss geometry is *least similar* to the average loss geometry of the cluster they belong to (recall Assumption 2.2.i).

(iii) The constant $\max_{j \in \mathcal{B}} \frac{1}{|\mathcal{B}|} \sum_i \frac{1}{c_{ij}}$ captures the user whose preference is *least similar* to the optimal service parameter of the cluster they belong to (recall Assumption 2.2.ii).

(iv) Even within a cluster all terms are averages, so a few poorly aligned pairs of users don't hurt the bound if the cluster sizes are large.

**Fair objective.** While minimizing the total loss is beneficial from the provider's point of view in keeping users satisfied on average, it is undesirable in human-centric applications if the provided services result in unfavorable or harmful outcomes towards some demographic groups.

Suppose the $n$ users come from $m$ different demographic groups ($m$ is typically small, say racial groups, gender). We denote the groups as $\mathcal{A} = \{\mathcal{A}_1, \ldots, \mathcal{A}_m\} \subset [n]$. The fairness objective is defined as the maximum average loss suffered by any group:

$$\Phi(\Theta, \mathcal{A}) = \max_{i \in [m]} \mathcal{L}(\Theta, \mathcal{A}_i)/|\mathcal{A}_i|. \tag{4}$$

[15] defined this objective in the context of fair $k$-means where the points and group identities are *known* and gave a *non-constructive* proof that if a $c-$approximate solution for $k$-means exists, it is

---

[1]In addition, as stated earlier, we tackle the lack of prior information on $\phi_i$ and the function form of $\mathcal{L}_i$—this challenge is particular to our setting and does not arise in related standard problems of clustering ($k$-means, $k$-mediods) or resource allocation (facility location problems).

$m \cdot c$-approximate for fair $k$-means. For the fairness objective, we slightly modify our algorithm, by simply reweighting the probability to select an user by the inverse of the size of their demographic group to result in Fair AcQUIre (Algorithm 2 in Appendix E).

**Theorem 3.2.** *Consider $n$ users with unknown preferences $\{\phi_1, \ldots, \phi_n\} \subset \mathbb{R}^d$, and associated loss functions $\mathcal{L}_i(\cdot, \cdot)$ satisfying Assumptions 2.1 and 2.2 with bandit access. Suppose these users belong to $m$ demographic groups $\mathcal{A} = \{\mathcal{A}_1, \ldots, \mathcal{A}_m\} \subset [n]$. Let $\Theta_{\mathrm{fair}} \subset \mathbb{R}^d$ be the set of $k$ services minimizing the fairness objective $\Phi$ given in (4). If Algorithm 2 is used to obtain $k$ services $\Theta_k$, then the following bound holds:*

$$\mathbb{E}_{\Theta_k}[\Phi(\Theta_k, \mathcal{A})] \leq m K_{\mathrm{fair}}(2 + \log k) \cdot \Phi(\Theta_{\mathrm{fair}}, \mathcal{A}),$$

*where the expectation is taken over the randomization of the algorithm and $K_{\mathrm{fair}}$ is defined in (10).*

## 4  Generalization in Linear Predictors

In practical settings, a provider would want to design services that not only keep the subscribed users satisfied but also attract new users to subscribe by generalizing the services to users it has never interacted with before. Now instead of considering $n$ users, suppose that each $i \in [N]$ represents a subpopulation with its own (sub-Gaussian) distribution of features, and the provider can interact with finite samples $n_i$ from these distributions. A question that arises is whether we can deal with this finite-sample-from-subpopulations scenario, and how does the number of samples affect the algorithm's output to unseen users. In this section, we answer this question for the special case of linear predictors (i.e., regression loss).

In this section we restrict ourselves to the special case of linear prediction tasks, where the goal is to accurately predict the score of a user as a linear function of their features. The score for a user in the $i^{\mathrm{th}}$ subpopulation with zero-mean random feature $x \in \mathbb{R}^d$ is generated as $y = \phi_i^\top x$ where both the true linear regressor $\phi_i \in \mathbb{R}^d$ and the feature covariance $\mathbb{E}_x[xx^\top] = \Sigma_i$ are unknown. Suppose a service uses a linear regressor $\theta \in \mathbb{R}^d$, to predict the score for this user as $\theta^\top x \in \mathbb{R}$. The loss for this subpopulation for this service is defined as the expected squared error between the predicted and actual scores, i.e., $\mathcal{L}_i(\theta, \phi_i) = \mathbb{E}_{(x,y)}[(\theta^\top x - y)^2] = \|\theta - \phi_i\|_{\Sigma_i}^2$.

**Assumption 4.1.** For subpopulation $i$, features are independent draws from a zero-mean sub-Gaussian distribution. For a random feature $x \in \mathbb{R}^d$ and for any $u \in \mathbb{R}^d$, such that $\|u\|_{\Sigma_i} = 1$, $u^\top x \in \mathbb{R}$ is sub-Gaussian with variance proxy $\sigma_i^2$.[2]

**Assumption 4.2.** We assume that the decision to choose between different services happens at a subpopulation level and not an individual level.

To illustrate Assumption 4, consider the example of a provider that offers personalized services to schools (subpopulations) such as online library resources wherein the service provider queries students about the experience. Each school typically has a considerable number of students, but only a subset of them may actively respond to such queries. Once a school selects the service, it is made available to all students, and the provider could implement a system where students are encouraged to fill out a feedback form after using their service.

Suppose only $n_i$ users from subpopulation $i$ are subscribed to the services. Thus, upon choosing a service parameterized by $\theta \in \mathbb{R}^d$ the provider observes an *empirical loss*, which is given by

$$\widehat{\mathcal{L}}_i(\theta, \phi_i) = \frac{1}{n_i} \sum_{j \in [n_i]} (\theta^\top x_i^j - y_i^j)^2,$$

where $\{(x_i^j, y_i^j)\}_{j \in [n_i]}$ are private unknown features and scores of the users. We stress that the service gets to see the value of the user loss function at the deployed $\theta$ (bandit feedback), but not the features of each subpopulation.

**Assumption 4.3.** The number of users from each subpopulation is greater than the dimension of the linear predictor, i.e. $n_i \geq d$ for all $i \in [N]$.

In this setting, given a set of services parameterized by $\Theta_{t-1} = \{\theta_1, \ldots, \theta_{t-1}\}$ the Steps 5-6 of Algorithm 1 proceed with these finite sample averages $\widehat{\mathcal{L}}_i(\Theta_{t-1}, \phi_i) = \min_{j \in [t-1]} \widehat{\mathcal{L}}_i(\theta_j, \phi_i)$. In

---

[2]A random variable $x$ is sub-Gaussian with variance proxy $\sigma^2$ if $\mathbb{E}[\exp(\lambda x)] \leq \exp(\frac{\sigma^2 \lambda^2}{2}) \ \forall \lambda \in \mathbb{R}$.

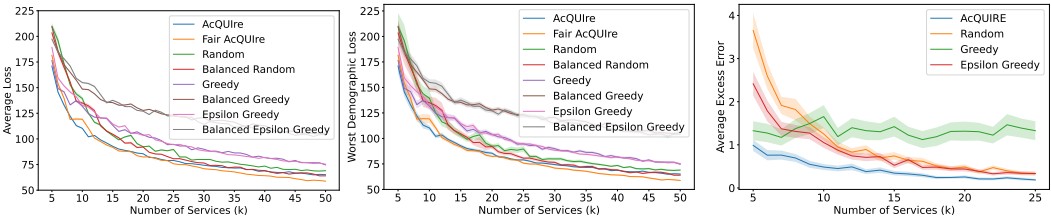

(a) Avg. loss (all Census groups)    (b) Avg. loss (worst demographic)    (c) Avg. excess error (ML10M task)

Figure 1: Fig. 1a and 1b show the performance of various user selection strategies on the travel time prediction task on the Census data. Notably, our findings reveal that the greedy and epsilon-greedy baselines exhibit strong performance for $k < 10$. However, as the value of $k$ grows, these strategies prove myopic, with random sampling surpassing their effectiveness. AcQUIre and Fair AcQUIre consistently emerge as the two best baselines for both tasks. Fig. 1c presents the average excess error for the movie recommendation task. Remarkably, the greedy algorithm demonstrates efficacy when $k$ is small. Epsilon-greedy, employing an explore-vs-exploit approach, successfully overcomes myopic tendencies. Nevertheless, AcQUIre continues to be the best baseline for data collection.

Step 7, multiple ways can be adopted by the provider to estimate $\phi_i$. Users can be given incentives to provide a batch of feature score pairs, or gradient free methods can be used to estimate the optimal solution to the regression problem. The generalization guarantee of Algorithm 1 to the total *expected loss* is stated below.

**Theorem 4.4.** *Suppose users belong to $N$ subpopulations satisfying Assumptions 4.1, 4.2, and 4.3. Let $\{n_i\}_{i \in [N]}$ denote the number of samples per subpopulation. Let $\Theta_k$ be the output of Algorithm 1 using finite sample estimates $\widehat{\mathcal{L}}_i(\cdot, \phi_i)$ and $\Theta_{\mathrm{OPT}}$ be the optimal solution of the expected loss. Then, for any $\epsilon \in (0, 1)$, if $n_i = \Omega(\frac{\sigma_i^4 \sqrt{N} \log(2/\delta)}{\epsilon^2})$ for all $i \in [N]$, the following inequality holds with probability at least $1 - \delta$:*

$$\mathbb{E}_{\Theta_k}[\mathcal{L}(\Theta_k, [N])] \leq \frac{1+\epsilon}{1-\epsilon} K_{\mathrm{OPT}}(2 + \log k)\mathcal{L}(\Theta_{\mathrm{OPT}}, [N]), \text{ where } K_{\mathrm{OPT}} \text{ is as defined in (3) and}$$

$$c_{ij} = \frac{1}{2} \min \left\{ \lambda_{\min}(\widehat{\Sigma}_i, \widehat{\Sigma}_j), \frac{1}{\lambda_{\min}(\widehat{\Sigma}_i, \widehat{\Sigma}_j)} \right\}, \text{ with } \widehat{\Sigma}_i = \frac{1}{n_i} \sum_{l \in [n_i]} x_i^l (x_i^l)^\top, \ \widehat{\Sigma}_j = \frac{1}{n_j} \sum_{l \in [n_j]} x_j^l (x_j^l)^\top.$$

The proof is presented in Appendix F; we provide a brief overview here. We apply the Chernoff bound to the difference between the *empirical loss* and *expected loss*. Note that $(i)$ even if the same set of services are provided, the loss minimizing service for the empirical loss may be different from the expected loss for any subpopulation, and $(ii)$ the optimal set of services for the total empirical loss and the total expected loss are different. Handling these carefully, and utilizing Theorem 3.1 concludes the proof.

*Remark* 4.5. Note that $\widehat{\Sigma}_i$ is the empirical feature covariance of subpopulation $i$. The term $c_{ij}$ captures the alignment between two subpopulation's loss geometry, and here is equal to half of the *minimum generalized eigenvalue* of the empirical feature covariances of the the respective subpopulations. This quantity is the largest constant satisfying Assumption 2.2 (cf. Appendix F).

## 5   Experiments

We empirically demonstrate[3] the benefits of our algorithm on a commute time prediction task based on 2021 US Census data[4] and a semi-synthetic movie recommendation task on the MovieLens10M dataset. In each task the multi-service provider has initially no access to data. Our goal is to evaluate the effectiveness of the iterative data collection and service initialization procedure in comparison to established baselines. Below we first describe both the tasks and then discuss the baselines we consider for our evaluation.

**Census Data.** We consider the task of predicting daily work commute times, based on 2021 US census data from FOLKTABLES [13]. We illustrate a potential use case: the *provider* is a transport authority offering *services* in the form of personalized podcasts. If the duration of a service is similar to the commute time of a user, that user will be able to consume the media while travelling to work. Hence an accurate prediction of the commute time may be useful in providing services tailored to the users.

---

[3]All our code is available at `https://anonymous.4open.science/r/MultiServiceInitialization-A422`

[4]`https://www.census.gov/programs-surveys/acs/data.html`.

The dataset has $N = 7025$ subpopulations defined by area zip codes. We use $k$ linear predictors (as services) with user features (education, income, transportation mode, age etc). Refer to Appendix G for details on data pre-processing. The features and commute times are *unknown a priori* to the provider. At time step $t \le k$, suppose that the provider offers a set of services parameterized by $\Theta_{t-1}$. The provider observes the losses (squared prediction error of their commute times as discussed in Section 4) across different subpopulations $\{\widehat{\mathcal{L}}_i(\Theta_{t-1}, \phi_i)\}_{i \in [N]}$. Then the provider selects a subpopulation $l \in [N]$, to get the user feature and commute time data and then estimates $\phi_l$ via least squares regression. This selection can be done via our proposed method or one of the baseline strategies discussed later. The provider then centers its next service on $\phi_l$ and updates the list of offered services $\Theta_t = \Theta_{t-1} \cup \phi_l$. Note that in this process the provider only observes features and commute times of users in the $k$ selected subpopulations and $k \ll N$. In Figure 2 we compare runtimes, and observe that even with 1 billion users, AcQUIre takes 300 sec, whereas the greedy and epsilon greedy methods take $> 10^5$ sec even for 10 million users. With 5000 services, AcQUIre takes $< 900$ secs, whereas the runtimes for greedy and epsilon greedy are in the range of $10^5$ secs.

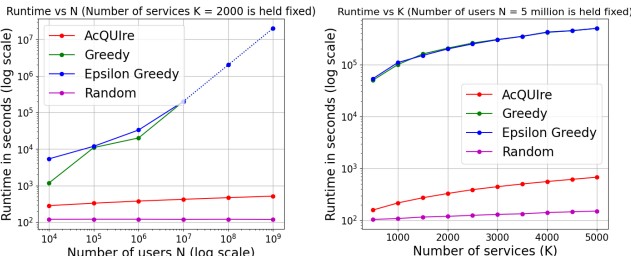

Figure 2: Runtimes for AcQUIre and baselines as number of users ($N$) and services ($K$) vary.

**Movie Recommendations.** We conduct a semi synthetic experiment based on the widely used Movielens10M data set [19] containing 10000054 ratings across 10681 movies by 71567 viewers. We hold out the top $m = 200$ movies and pre-process the data set to divide viewers into $N = 1000$ user subpopulations based on similarity of their ratings on the remaining movies (cf. Appendix G). Our goal is to evaluate the generalization performance of the baselines to viewers that the provider has never interacted with before. Thus during the service initialization phase, only half of the users in each of the $N$ subpopulations interact with the provider (train set), and we evaluate the performance of the algorithm by the loss incurred by the initialized services on data of the other half of the users that no prior interaction with the provider (test set). For subpopulation $i \in [N]$, the solution to the following optimization problem denotes the user and item embedding:

$$(U_i, \phi_i) = \arg\min_{(U, \phi) \in \mathbb{R}^{n_i \times d} \times \mathbb{R}^{d \times m}} \|(U\phi - r_i)_{\Omega_i^{\text{train}}}\|_2^2, \tag{5}$$

where $r_i$ is the true user ratings and $\Omega_i^{\text{train}}$ is the list of movies rated by the users in subpopulation $i$. Since $r_i$ is a sparse matrix we consider the prediction error only on the movies they rated, i.e. $\Omega_i^{\text{train}}$. User loss / dissatisfaction for a recommendation model, parameterized by $\theta \in \mathbb{R}^{d \times m}$, is captured by the excess error, namely

$$\mathcal{L}_i(\theta, \phi_i) = \|(U_i\theta - r_i)_{\Omega_i^{\text{train}}}\|_2^2 - \|(U_i\phi_i - r_i)_{\Omega_i^{\text{train}}}\|_2^2. \tag{6}$$

This value typically indicates how unhappy users are with the suggested movies with respect to their preferred movies. The provider initially doesn't know the user ratings. At time step $t \le k$, suppose the provider offers a set of recommendation models $\Theta_{t-1}$. Users choose the service with the best recommendations and the provider observes the losses across different subpopulations $\{\mathcal{L}_i(\Theta_{t-1}, \phi_i)\}_{i \in [N]}$. Then the provider selects a subpopulation $l \in [N]$ to estimate $\phi_l$ via (5). This selection can be done via our proposed method or one of the baseline strategies discussed below. The provider then centers its next model on $\phi_l$ and updates the list of offered models $\Theta_t = \Theta_{t-1} \cup \phi_l$. In this process the provider only observes movie ratings of the users in the $k$ selected subpopulations. Once the services are initialized we evaluate the performance on the movies rated in the test set denoted by $\{\Omega_1^{\text{test}}, \ldots, \Omega_N^{\text{test}}\}$.

**Baselines.** Both our tasks iterate through the steps of observing *User Behavior*, *User Selection* to gather data, designing *New Service* to update set of offered services. Through our experiments we wish to empirically evaluate different **User Selection** strategies with respect to AcQUIre (line

6 in Algorithm 1). The different user selection strategies result in the following baselines: (i) Random: $P(l = i) = 1/n$, (ii) Greedy: $l = \text{argmax}_{i \in [n]} \mathcal{L}_i(\Theta_{t-1}, \phi_i)$, (iii) Epsilon Greedy: $l = \text{argmax}_{i \in [n]}(\mathcal{L}_i(\Theta_{t-1}, \phi_i) + \epsilon_i)$ where $\epsilon_1, \ldots, \epsilon_n$ denotes zero mean i.i.d. noise. Given that the Census Dataset comprises various racial demographic groups of varying sizes (with the smallest group being ten times smaller than the largest group), and considering our interest in the fairness objective (4), we explore incorporating these size imbalances into our algorithms. Consequently, we introduce three additional baselines, wherein the selection criteria are scaled by the corresponding group sizes: (iv) Balanced Random, (v) Balanced Greedy, and (vi) Balanced Epsilon Greedy. We benchmark them against Fair AcQUIre (Algorithm 2) which has guarantees as as stated in Theorem 3.2.

**Evaluation:** Each algorithm is run for 500 initialization seeds, the averages are reported in Figure 1.

**Runtimes:** We compare the runtimes of AcQUIre with the baselines, and study the affect of the number of users ($N$) and number of services ($K$) in Figure 2. We find the runtimes of AcQUIre to be of the similar order of magnitude of random initialization, meanwhile performing much better than random, and the much slower greedy and epislon greedy initialization schemes.

**Impact of Initialization:** Once a set of services are initialized, with more user interactions, the provider updates the services on new data to improve the quality (indicated by the reduction in total loss). To evaluate the importance of initialization, we conducted experiments using two different optimization algorithms: (i) Generalized k-means: The services are iteratively updated by training each service on the current group of subpopulations selecting it. After updating the service parameters, the subpopulations reselect their best service. This process repeats until convergence. (ii) Multiplicative weights update [12]: Similar to k-means, but each subpopulation can have users choosing different services simultaneously. Both generalized k-means and the multiplicative weights update guarantee that the total loss reduces over time [12].

In our experiments, we initialize a set of services using AcQUIre and other baseline methods, then let both optimization algorithms run until convergence. We plot the total loss versus the number of iterations (Figure 3). Our results demonstrate that AcQUIre leads to: (1) faster convergence, and (2) lower final loss (initializing with AcQUIre converges to lower losses; other initialization schemes are prone to being stuck in suboptimal local minima). These findings highlight the significance of a robust initialization strategy. By starting with a better initial configuration, the optimization algorithms can more effectively reach higher quality solutions.

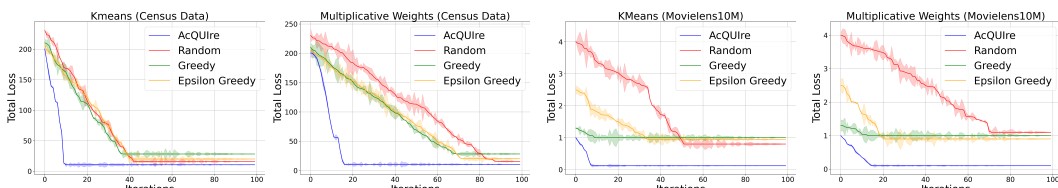

Figure 3: We study the importance of initialization in both the convergence rate and quality of converged solution of optimization algorithms. We find AcQUIre converges both faster and to a lower total loss across optimization methods (kmeans and multiplicative weights) as well as datasets.

## 6 Conclusion

We study the problem of initializing services for a provider catering to a user base with diverse preferences. We address the challenges of unknown user preferences, only bandit (zeroth-order) feedback from the losses, and the non-convexity of the optimization problem, by proposing an algorithm that designs services by adaptively querying data from a small set of users. We also consider the fairness aspect of such design in human centric applications. Our proposed algorithm has theoretical guarantees on both the average and fair loss objectives. There are open questions relating to quantifying the robustness of the proposed initialization algorithm to noisy observations, perturbations, or outliers in the finite sample case when the feature distribution is heavy-tailed, which is a direction for future work.

## Acknowledgements

Dean, Fazel, Morgenstern and Ratliff were supported in part by NSF CCF-AF 2312774/2312775. Additionally, Morgenstern's work was supported by NSF CCF 2045402. Dean's research was supported by a gift from Wayfair, a LinkedIn research award, and NSF OAC 2311521. Ratliff's research was supported by NSF CNS 1844729 and Office of Naval Research YIP Award N000142012571. Fazel was supported in part by awards NSF TRIPODS II 2023166, CCF 2007036, CCF 2212261.

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

## A  More Discussion on Related Works

**Retention.** User retention in machine learning systems is closely related to the decision dynamics between the provider and users studied in the single service setting by Hashimoto et al. [20], Zhang et al. [41] and multiple services setting by Dean et al. [12], Ginart et al. [18]. In settings with multiple sub-populations of users of different types, the question of retention has been explored in parallel with the issue of fairness. These works typically focus on the stability of the dynamics at equilibrium. However, the final outcome is heavily influenced by the initial data the provider has and the initial configuration (partitioning) of the users across the offered services. Our work addresses *the impact of initialization on the final outcome with theoretical guarantees.*

**Mixture of Experts.** This model specialization is also explored in the mixture of experts literature (see, e.g., [31]), which uses multiple 'expert' models to enhance accuracy and robustness, assigning inputs to the most suitable expert based on their features through a gating mechanism.

**Clustering.** [29] improved the analysis of [5] to show an approximation ratio $5(2 + \log k)$ and showed a family of instances with $5 \log k$ approximation ratio, thus showing $k$-means++ is tight. Recent works in clustering [24, 2, 26] provide a constant approximation ratio, and although they are polynomial time algorithms, these methods are data inefficient and rely on *knowing* the points *a priori.*

**Facility Location Problem.** Our algorithm has some resemblances with the facility location problem where there are $n$ users, and a provider can set up at most $k$ facilities at one of $m$ candidate locations (also known as the $k$-medoids objective [4]). One key difference is the provider can choose from $m$ predecided locations, compared to our setting where the optimization space is infinite. However our algorithm initializes only at $k$ of the $n$ user preferences, hence it can be viewed as the candidate locations simply being the user preferences and thus $m = n$. A typical greedy algorithm for the $k$-medoids objective proceeds by evaluating the marginal decrement in total loss for *all possible candidates* and selects the candidate with maximal loss reduction. Thus (i) the algorithm would need to know all $\{\phi_1, \ldots, \phi_n\}$ apriori, and (ii) deploy $n$ services to obtain all $n^2$ function evaluations $\mathcal{L}_j(\phi_i, \phi_j), \ \forall i, j \in [n]$. This is infeasible in applications like online recommendation systems where $n$ is very large and typically a provider has the capacity to deploy only $k \ll n$ services. Hence a $k$-medoids like objective will not be reasonable in our setup with incomplete information.

[6] studied the case where the provider had prior access to only $N < n$ users' utility functions before deploying services and the goal was to minimize the worst case total (over all $n$ users) loss. However their results assume they can solve a computationally hard mixed integer program optimally.

**Preference Learning.** Given a *fixed* set of items or services, [38] focus on learning preference distributions of users. [8] extend this to the setting where the user losses are given by an identical metric, and they learn both preferences and the metric efficiently. Our setting focuses on the *design of services rather than learning preferences over a fixed set of services, and we also allow each user to have a different loss function and loss geometry.*

## B  Motivating Example

We conceptualize 'services' broadly, encompassing both sets of independent learners—such as various service providers collaborating on initialization—and single learners with multiple models, like companies with diverse platforms or multi-model servers in federated learning settings. This initialization process is further relevant for social planners aiming to facilitate the coordination of service initialization.

**Netflix Example.** All details are borrowed from the actual working of Netflix (refer `https://recoai.net/netflix-recommendation-system-how-it-works/` for a detailed description and more references). The Netflix homepage displays several rows of suggestions. Each row is a collection of movies and the rows are arranged from top to bottom in decreasing order of likelihood that the user will pick movies to watch. The user has a complete choice of which row to select a movie to watch from. The different rows are generated by different underlying recommendation models parametrized by $\theta_1, \ldots, \theta_K$. Our model abstracts these rows out as services. Netflix maintains $K = 1300$ such different recommendation models. Users' decision of watching a movie from the available rows of recommendations on the Netflix homepage informs the provider (Netflix) which of

the rows (parameterized recommendation model) the user prefers most. This let's them finetune their recommendation models.

## C  Mappings satisfying Assumptions 2.1 and 2.2

In this appendix, we provide more details on examples that satisfy Assumptions 2.1 and 2.2. For the **squared error loss for linear predictors**, we refer the reader to Lemma F.1 for a derivation of $c_{ij}$. The remainder of the example classes are detailed below.

### C.1  Huber Loss

Recall that the Huber loss is defined by

$$\mathcal{L}_i(\theta, \phi_i) = \begin{cases} \frac{1}{2}\|\theta - \phi_i\|^2 & \text{if } \|\theta - \phi_i\| \leq \delta, \\ \delta(\|\theta - \phi_i\| - \frac{1}{2}\delta) & \text{otherwise.} \end{cases}$$

Note that the Huber loss varies quadratically when the error $\|\theta - \phi_i\| \leq \delta$ and linearly otherwise. We will refer to these as quadratic and linear regime of the Huber loss subsequently.

We divide the derivation into 2 subcases.

**Case 1.** $\|\theta - \phi_i\| \leq 2\delta$. Within **Case 1**, there are several subcases to consider. We analyze each one separately.

- **At least one of the terms on the right is linear:** Under the condition of **Case 1**, $\mathcal{L}_i(\theta, \phi_i) \leq \frac{3}{2}\delta^2$. Note that the Huber loss is monotonicly increasing in the prediction error. The function value in the linear in $\|\theta - \phi_i\|$ is at least $\frac{1}{2}\delta^2$. Thus, if one of the terms in $\mathcal{L}_j(\theta, \phi_j) + \mathcal{L}_j(\phi_i, \phi_j)$ were in the linear regime, the $\mathcal{L}_j(\theta, \phi_j) + \mathcal{L}_j(\phi_i, \phi_j) \geq \frac{1}{2}\delta^2$ and $\frac{1}{3}\mathcal{L}_i(\theta, \phi_i) \leq \mathcal{L}_j(\theta, \phi_j) + \mathcal{L}_j(\phi_i, \phi_j)$.

- **Both terms on the right are quadratic:** In this sub-case, it needs to be shown when both terms of $\mathcal{L}_j(\theta, \phi_j) + \mathcal{L}_j(\phi_i, \phi_j)$ are in the quadratic regime the bound still holds.

  By the triangle inequality on norms, we have that

$$\|\theta - \phi_i\| \leq \|\theta - \phi_j\| + \|\phi_i - \phi_j\|.$$

Using the power mean inequality, namely

$$a \leq b + c \implies a^2 \leq 2(b^2 + c^2),$$

on the triangle inequality, we deduce the following implication:

$$\|\theta - \phi_i\| \leq \|\theta - \phi_j\| + \|\phi_i - \phi_j\| \implies \frac{1}{2}\|\theta - \phi_i\|^2 \leq \|\theta - \phi_j\|^2 + \|\phi_i - \phi_j\|^2.$$

If the $\mathcal{L}_i(\theta, \phi_i)$ is quadratic, we have that $\frac{1}{2}\mathcal{L}_i(\theta, \phi_i) \leq \mathcal{L}_j(\theta, \phi_j) + \mathcal{L}_j(\phi_i, \phi_j)$. Now, let $\mathcal{L}_i(\theta, \phi_i)$ be linear. Then, we deduce that

$$\mathcal{L}_j(\theta, \phi_j) + \mathcal{L}_j(\phi_i, \phi_j) = \frac{1}{2}(\|\theta - \phi_j\|^2 + \|\phi_i - \phi_j\|^2) \geq \frac{1}{4}\|\theta - \phi_i\|^2.$$

Since the equation $\frac{1}{4}x^2 - \frac{1}{3}x + \frac{1}{6} = 0$ has no real roots, the expression is always positive. Hence, by substituting $x = \frac{\|\theta - \phi_i\|}{\delta}$, we have that

$$\frac{1}{4}\|\theta - \phi_i\|^2 \geq \frac{1}{3}\delta(\|\theta - \phi_i\| - \frac{1}{2}\delta) = \frac{1}{3}\mathcal{L}_i(\theta, \phi_i).$$

Therefore, we deduce that

$$\frac{1}{3}\mathcal{L}_i(\theta, \phi_i) \leq \mathcal{L}_j(\theta, \phi_j) + \mathcal{L}_j(\phi_i, \phi_j).$$

**Case 2.** $\|\theta - \phi_i\| > 2\delta$. First, observe that

$$
\begin{aligned}
\frac{1}{3}\mathcal{L}_i(\theta, \phi_i) &= \frac{1}{3}\delta(\|\theta - \phi_i\| - \frac{1}{2}\delta) \\
&= \frac{1}{3}\delta\|\theta - \phi_i\| + \frac{1}{6}\delta\|\theta - \phi_i\| - \frac{1}{6}\delta\|\theta - \phi_i\| - \frac{1}{6}\delta^2 \\
&\leq \frac{1}{3}\delta\|\theta - \phi_i\| + \frac{1}{6}\delta\|\theta - \phi_i\| - \frac{1}{3}\delta^2 - \frac{1}{6}\delta^2 \\
&\leq \frac{1}{2}\delta\|\theta - \phi_i\| - \frac{1}{2}\delta^2,
\end{aligned}
$$

where the second to last inequality follows from the fact that $\|\theta - \phi_i\| > 2\delta$. By the triangle inequality on norms, we have that

$$
\|\theta - \phi_i\| \leq \|\theta - \phi_j\| + \|\phi_i - \phi_j\|.
$$

In turn, this implies either $\|\theta - \phi_j\|$ or $\|\phi_i - \phi_j\|$ is greater than $\frac{1}{2}\|\theta - \phi_i\|$. Without loss of generality, assume $\|\theta - \phi_j\| \geq \frac{1}{2}\|\theta - \phi_i\| \geq \delta$. Thus

$$
L_j(\theta, \phi_j) = \delta(\|\theta - \phi_j\| - \frac{1}{2}\delta) \geq \frac{1}{2}\delta\|\theta - \phi_i\| - \frac{1}{2}\delta^2 \geq \frac{1}{3}\mathcal{L}_i(\theta, \phi_i).
$$

Since $\mathcal{L}_j(\phi_i, \phi_j) \geq 0$, we have that

$$
\frac{1}{3}\mathcal{L}_i(\theta, \phi_i) \leq \mathcal{L}_j(\theta, \phi_j) + \mathcal{L}_j(\phi_i, \phi_j).
$$

## C.2 The normalized cosine distance

Recall that the normalized cosine distance is given by

$$
\mathcal{L}_i(\theta, \phi_i) = 1 - \theta^\top \phi_i \quad \text{where} \quad \|\theta\|_2 = \|\phi_i\|_2 = 1.
$$

Therefore, we have that

$$
1 - \theta^\top \phi_i = \frac{1}{2}(\|\theta\|_2^2 + \|\phi_i\|_2^2 - 2\theta^\top \phi_i) = \frac{1}{2}\|\theta - \phi\|_2^2.
$$

Assumption 2.2 is satisfied with $c_{ij} = \frac{1}{2}$ by using triangle inequality followed by power mean inequality.

## C.3 Mahalanobis distance

Consider the Mahalanobis distance which is defined by

$$
\mathcal{L}_i(\theta, \phi_i) = \|\theta - \phi_i\|_{\Sigma_i},
$$

where $\Sigma_i$ is full rank and $c_{ij} = \min\{\lambda_{\min}(\Sigma_i, \Sigma_j), \frac{1}{\lambda_{\min}(\Sigma_i, \Sigma_j)}\}$. The derivation is similar to proof of Lemma F.1.

# D Proof of Theorem 3.1

We note that the parameters of the services initialized by Algorithm 1 are a subset of the users' unknown preferences. This allows us to define the notion of *covering*.

**Definition D.1.** Let $\Theta \subset \{\phi_1, \ldots, \phi_n\}$ be a set of services. A cluster $\mathcal{B} \in \mathcal{C}(\Theta_{\mathrm{OPT}})$ is said to be *covered* by $\Theta$ if there exists $i \in \mathcal{B}$ such that $\phi_i \in \Theta$. If no such $i \in \mathcal{B}$ exists, then the cluster $\mathcal{B}$ is said to be *uncovered*.

The proof idea from here on is to show that there exists an approximation ratio $K_{\mathrm{OPT}}$ for the *covered* clusters, and is shown in Lemma D.2 and D.3.

**Lemma D.2.** *Let $\mathcal{B} \in \mathcal{C}(\Theta_{\mathrm{OPT}})$ be an arbitrary cluster in $\mathcal{C}(\Theta_{\mathrm{OPT}})$, and let $\theta \in \mathbb{R}^d$ be a service centered on the preference of a user $j$ chosen uniformly at random from $[n]$. The expected loss of users in $\mathcal{B}$, conditioned on $j \in \mathcal{B}$, satisfies*

$$\mathbb{E}_\theta[\mathcal{L}(\{\theta = \phi_j\}, \mathcal{B}) \mid j \in \mathcal{B}] \leq \left( \max_{j \in \mathcal{B}} \frac{2}{|\mathcal{B}|} \sum_i \frac{1}{c_{ij}} \right) \mathcal{L}(\Theta_{\mathrm{OPT}}, \mathcal{B}).$$

*Proof.* Since we choose a user from $\mathcal{B}$, the conditional probability that we choose some fixed $\phi_j$ as the parameter for the service is precisely $P(\theta = \phi_j \mid \theta \in \mathcal{B}) = \frac{1}{|\mathcal{B}|}$. Let $\mathcal{J}(\mathcal{B})$ denote the best service covering all the points in $\mathcal{B}$, and then compute

$$\mathbb{E}_\theta[\mathcal{L}(\{\theta = \phi_j\}, \mathcal{B}) \mid j \in \mathcal{B}] = \sum_{j \in \mathcal{B}} P(\theta = \phi_j)\mathcal{L}(\theta, \mathcal{B}) = \sum_{j \in \mathcal{B}} \frac{1}{|\mathcal{B}|} \sum_{i \in \mathcal{B}} \mathcal{L}_i(\phi_j, \phi_i).$$

Using Assumption 2.2.(ii) with $\theta = \mathcal{J}(\mathcal{B})$, we have that

$$\mathbb{E}_\theta[\mathcal{L}(\{\theta = \phi_j\}, \mathcal{B}) \mid j \in \mathcal{B}] \leq \sum_{j \in \mathcal{B}} \sum_{i \in \mathcal{B}} \frac{1}{|\mathcal{B}|} \left( \frac{1}{c_{ij}} \left( \mathcal{L}_j(\mathcal{J}(\mathcal{B}), \phi_j) + \mathcal{L}_i(\mathcal{J}(\mathcal{B}), \phi_i) \right) \right),$$

$$= \sum_{j \in \mathcal{B}} \left( \frac{1}{|\mathcal{B}|} \sum_{i \in \mathcal{B}} \frac{1}{c_{ij}} \right) \mathcal{L}_j(\mathcal{J}(\mathcal{B}), \phi_j) + \sum_{i \in \mathcal{B}} \left( \frac{1}{|\mathcal{B}|} \sum_{j \in \mathcal{B}} \frac{1}{c_{ij}} \right) \mathcal{L}_i(\mathcal{J}(\mathcal{B}), \phi_i).$$

Noting that $c_{ij} = c_{ji}$, by swapping the indices of the second term in the above summand, the second term is identical to the first term. Therefore, we deduce that

$$\mathbb{E}_\theta[\mathcal{L}(\{\theta = \phi_j\}, \mathcal{B}) \mid j \in \mathcal{B}] \leq 2 \sum_{j \in \mathcal{B}} \left( \left( \frac{1}{|\mathcal{B}|} \sum_{i \in \mathcal{B}} \frac{1}{c_{ij}} \right) \mathcal{L}_j(\mathcal{J}(\mathcal{B}), \phi_j) \right).$$

Now, using the fact that $\max_{j \in \mathcal{B}} \left( \frac{1}{|\mathcal{B}|} \sum_{i \in \mathcal{B}} \frac{1}{c_{ij}} \right)$ is a uniform upper bound for the multiplier in the above expression, we bring it outside the summation as a constant. Therefore, we deduce that

$$\mathbb{E}_\theta[\mathcal{L}(\{\theta = \phi_j\}, \mathcal{B}) \mid j \in \mathcal{B}] \leq \left( \max_{j \in \mathcal{B}} \frac{2}{|\mathcal{B}|} \sum_i \frac{1}{c_{ij}} \right) \mathcal{L}(\mathcal{J}(\mathcal{B}), \mathcal{B}), \tag{7}$$

where we have used the fact that $\mathcal{L}(\mathcal{J}(\mathcal{B}), \mathcal{B}) = \sum_{i \in \mathcal{B}} \mathcal{L}_i(\mathcal{J}(\mathcal{B}), \phi_i)$. Since $\mathcal{B} \in \mathcal{C}(\Theta_{\mathrm{OPT}})$, the covering $\mathcal{J}(\mathcal{B})$ is the loss minimizing service among all services in $\Theta_{\mathrm{OPT}}$ for all points in $\mathcal{B}$. Thus, we have that

$$\mathbb{E}_\theta[\mathcal{L}(\{\theta = \phi_j\}, \mathcal{B}) \mid j \in \mathcal{B}] \leq \left( \max_{j \in \mathcal{B}} \frac{2}{|\mathcal{B}|} \sum_i \frac{1}{c_{ij}} \right) \mathcal{L}(\Theta_{\mathrm{OPT}}, \mathcal{B}).$$

This concludes the proof. $\qquad\square$

**Lemma D.3.** *Let $\mathcal{B} \in \mathcal{C}(\Theta_{\mathrm{OPT}})$ be an arbitrary cluster in $\mathcal{C}(\Theta_{\mathrm{OPT}})$, and let $\Theta_t \subset \mathbb{R}^d$ denote the parameters for a set of preexisting $t$ arbitrary services. Consider a new clustering $\Theta_{t+1} = \Theta_t \cup \theta$ where $\theta = \phi_j$ is a random service centered on user $j \in [n]$ selected with probability $P(\theta = \phi_j) \propto \mathcal{L}_j(\Theta_t, \phi_j)$. Then, the expected loss of $\mathcal{B}$, conditioned on $j \in \mathcal{B}$, satisfies*

$$\mathbb{E}_\theta[\mathcal{L}(\Theta_t \cup (\theta = \phi_j), \mathcal{B}) \mid j \in \mathcal{B}] \leq \frac{1}{\min_{j \in \mathcal{B}} \frac{1}{|\mathcal{B}|} \sum_{i \in \mathcal{B}} c_{ji}} \left( \max_{j \in \mathcal{B}} \frac{2}{|\mathcal{B}|} \sum_i \frac{1}{c_{ij}} \right) \mathcal{L}(\Theta_{\mathrm{OPT}}, \mathcal{B}).$$

*Proof.* Given that we are choosing a user from $\mathcal{B}$, the conditional probability that we center the new service on some fixed $\phi_j$ is precisely $\mathcal{L}(\Theta_t, \phi_j)/(\sum_{i \in \mathcal{B}} \mathcal{L}(\Theta_t, \phi_i))$. After adding $\phi_j$ to the list of services, a user $i$ will have loss $\min\{\mathcal{L}_i(\Theta_t, \phi_i), \mathcal{L}_i(\phi_j, \phi_i)\}$. Therefore we deduce that

$$\mathbb{E}_\theta[\mathcal{L}(\Theta_t \cup (\theta = \phi_j), \mathcal{B}) \mid j \in \mathcal{B}] = \sum_{j \in \mathcal{B}} P(\theta = \phi_j \mid \theta \in \mathcal{B}) \sum_{i \in \mathcal{B}} \mathcal{L}_i(\Theta_t \cup \theta, \phi_i),$$

$$= \sum_{j \in \mathcal{B}} \frac{\mathcal{L}_j(\Theta_t, \phi_j)}{\sum_{l \in \mathcal{B}} \mathcal{L}_l(\Theta_t, \phi_l)} \sum_{i \in \mathcal{B}} \min\{\mathcal{L}_i(\Theta_t, \phi_i), \mathcal{L}_i(\phi_j, \phi_i)\}. \tag{8}$$

By Assumption 2.2.$(i)$, for any $\theta \in \mathbb{R}^d$ we have that

$$c_{ji}\mathcal{L}_j(\theta, \phi_j) \leq \mathcal{L}_i(\theta, \phi_i) + \mathcal{L}_i(\phi_j, \phi_i).$$

We utilize the fact that given two functions $f, g : \Theta \to \mathbb{R}$, if $f(\theta) \leq g(\theta) \ \forall \theta \in \Theta$, then $\min_{\theta \in \Theta} f(\theta) \leq \min_{\theta \in \Theta} g(\theta)$. Hence, the following implication holds:

$$c_{ji} \min_{\theta \in \Theta} \mathcal{L}_j(\theta, \phi_j) \leq \min_{\theta \in \Theta} \mathcal{L}_i(\theta, \phi_i) + \mathcal{L}_i(\phi_j, \phi_i) \implies c_{ji}\mathcal{L}_j(\Theta_t, \phi_j) \leq \mathcal{L}_i(\Theta_t, \phi_i) + \mathcal{L}_i(\phi_j, \phi_i).$$

Summing over all $i \in \mathcal{B}$, we get that

$$\mathcal{L}_j(\Theta_t, \phi_j) \leq \frac{1}{\sum_{i \in \mathcal{B}} c_{ji}} \left( \sum_{i \in \mathcal{B}} \mathcal{L}_i(\Theta_t, \phi_i) + \mathcal{L}_i(\phi_j, \phi_i) \right)$$

Applying this to $\mathcal{L}_j(\Theta_t, \phi_j)$ on the right hand side of (8), we have that

$$\mathbb{E}_\theta[\mathcal{L}(\Theta_t \cup (\theta = \phi_j), \mathcal{B}) \mid j \in \mathcal{B}] \leq \sum_{j \in \mathcal{B}} \frac{1}{\sum_{i \in \mathcal{B}} c_{ji}} \left( 1 + \frac{\sum_{i \in \mathcal{B}} \mathcal{L}_i(\phi_j, \phi_i)}{\sum_{l \in \mathcal{B}} \mathcal{L}_l(\Theta_t, \phi_l)} \right) \sum_{i \in \mathcal{B}} \min\{\mathcal{L}_i(\Theta_t, \phi_i), \mathcal{L}_i(\phi_j, \phi_i)\}.$$

Since $\min\{a, b\} \leq a$, we further deduce that

$$\mathbb{E}_\theta[\mathcal{L}(\Theta_t \cup (\theta = \phi_j), \mathcal{B}) \mid j \in \mathcal{B}] \leq \sum_{j \in \mathcal{B}} \frac{1}{\sum_{i \in \mathcal{B}} c_{ji}} \sum_{i \in \mathcal{B}} \mathcal{L}_i(\phi_j, \phi_i)$$

$$+ \sum_{j \in \mathcal{B}} \frac{1}{\sum_{i \in \mathcal{B}} c_{ji}} \frac{\sum_{i \in \mathcal{B}} \mathcal{L}_i(\phi_j, \phi_i)}{\sum_{l \in \mathcal{B}} \mathcal{L}_l(\Theta_t, \phi_l)} \sum_{i \in \mathcal{B}} \mathcal{L}(\Theta_t, \phi_i),$$

$$= \sum_{j \in \mathcal{B}} \frac{2}{\sum_{i \in \mathcal{B}} c_{ij}} \sum_{i \in \mathcal{B}} \mathcal{L}_i(\phi_j, \phi_i).$$

Now, we use the upper bound $\frac{1}{\min_{j \in \mathcal{B}} \sum_{i \in \mathcal{B}} c_{ji}}$ for the multiplier to get

$$\mathbb{E}_\theta[\mathcal{L}(\Theta_t \cup (\theta = \phi_j), \mathcal{B}) \mid j \in \mathcal{B}] \leq \frac{2}{\frac{1}{|\mathcal{B}|} \min_{j \in \mathcal{B}} \sum_{i \in \mathcal{B}} c_{ji}} \sum_{j \in \mathcal{B}} \sum_{i \in \mathcal{B}} \frac{1}{|\mathcal{B}|} \mathcal{L}_i(\phi_j, \phi_i).$$

Note that $\sum_{j \in \mathcal{B}} \sum_{i \in \mathcal{B}} \frac{1}{|\mathcal{B}|} \mathcal{L}_i(\phi_j, \phi_i)$ is essentially the loss on choosing users from $\mathcal{B}$ uniformly randomly and centering the service on the chosen user. Plugging in the expression in (7) from the proof of Lemma D.2, we have that

$$\mathbb{E}_\theta[\mathcal{L}(\Theta_t \cup (\theta = \phi_j), \mathcal{B}) \mid j \in \mathcal{B}] \leq \frac{4}{\min_{j \in \mathcal{B}} \frac{1}{|\mathcal{B}|} \sum_{i \in \mathcal{B}} c_{ji}} \left( \max_{j \in \mathcal{B}} \frac{1}{|\mathcal{B}|} \sum_{i \in \mathcal{B}} \frac{1}{c_{ij}} \right) \mathcal{L}(\mathcal{J}(\mathcal{B}), \mathcal{B}). \quad (9)$$

Since $\mathcal{B} \in \mathcal{C}(\Theta_{\mathrm{OPT}})$, the covering $\mathcal{J}(\mathcal{B})$ is the loss minimizing service among all available services in $\Theta_{\mathrm{OPT}}$ for all points in $\mathcal{B}$. Therefore, we deduce that

$$\mathbb{E}_\theta[\mathcal{L}(\Theta_t \cup (\theta = \phi_j), \mathcal{B}) \mid j \in \mathcal{B}] \leq \frac{4}{\min_{j \in \mathcal{B}} \frac{1}{|\mathcal{B}|} \sum_{i \in \mathcal{B}} c_{ji}} \left( \max_{j \in \mathcal{B}} \frac{1}{|\mathcal{B}|} \sum_{i \in \mathcal{B}} \frac{1}{c_{ij}} \right) \mathcal{L}(\Theta_{\mathrm{OPT}}, \mathcal{B}),$$

which concludes the proof. $\qquad\square$

The following Lemma is an induction relating the losses on *covered* and *uncovered* clusters.

**Lemma D.4.** *Let $\Theta_t \subset \mathbb{R}^d$ denote the parameters of a set of $t$ arbitrary services. Consider $u > 0$ uncovered clusters from $\mathcal{C}(\Theta_{\mathrm{OPT}})$, denoted by $\mathcal{U}_t$, and let $\mathcal{H}_t$ denote the covered clusters. Suppose we add $v \leq u$ random services to $\Theta_t$, chosen with probability proportional to their current loss—i.e., $\mathcal{L}(\Theta_t, \phi_j)$—and let $\Theta_{t+v}$ denote the the resulting set of services. The following estimate holds:*

$$\mathbb{E}_{\Theta_{t+v}}[\mathcal{L}(\Theta_{t+v}, [n])] \leq (\mathbb{E}_{\Theta_t}[\mathcal{L}(\Theta_t, \mathcal{H}_t)] + K_{\mathrm{OPT}} \cdot \mathcal{L}(\Theta_{\mathrm{OPT}}, \mathcal{U}_t)) \cdot (1 + S_v) + \frac{u - v}{u} \cdot \mathbb{E}_{\Theta_t}[\mathcal{L}(\Theta_t, \mathcal{U}_t)],$$

*where $K_{\mathrm{OPT}}$ is as defined in (3) and $S_v = \left(1 + \frac{1}{2} + \ldots + \frac{1}{v}\right)$ is the harmonic series.*

*Proof.* Replacing the factor 8 in Lemma 3.3 from [5] by $K_{\text{OPT}}$ gives us the desired result.

$\square$

With the preceding technical lemmas, we are now ready to prove the main theorem which we restate for convenience.

**Theorem 3.1.** *Consider $n$ users with unknown preferences $\{\phi_1, \ldots, \phi_n\} \subset \mathbb{R}^d$, and associated loss functions $\mathcal{L}_i(\cdot, \cdot)$ satisfying Assumptions 2.1 and 2.2, with bandit access. Let $\Theta_{\text{OPT}} \subset \mathbb{R}^d$ be the set of $k$ services minimizing the total loss and $\mathcal{C}(\Theta_{\text{OPT}})$ the resulting partitioning of users (Definition 2.3). If Algorithm 1 is used to obtain $k$ services $\Theta_k$, then the following bound holds:*

$$\mathbb{E}_{\Theta_k}[\mathcal{L}(\Theta_k, [n])] \leq K_{\text{OPT}}(2 + \log k) \cdot \mathcal{L}(\Theta_{\text{OPT}}, [n]),$$

*where the expectation is taken over the randomization of the algorithm and $K_{\text{OPT}}$ is equal to*

$$\max_{\mathcal{B} \in \mathcal{C}(\Theta_{\text{OPT}})} \frac{4}{\min_{j \in \mathcal{B}} \sum_{i \in \mathcal{B}} c_{ij}} \left( \max_{j \in \mathcal{B}} \sum_{i \in \mathcal{B}} \frac{1}{c_{ij}} \right). \tag{3}$$

*Proof.* Consider $t = 1$, and $\mathcal{B} \in \mathcal{C}(\Theta_{\text{OPT}})$, the cluster in which the first chosen user belongs. Applying Lemma D.4 with $v = u = k - 1$ and the fact that $\mathcal{B}$ is the only covered cluster, we have that

$$\mathbb{E}_{\Theta_k}[\mathcal{L}(\Theta_k, [n])] \leq (\mathbb{E}_{\Theta_1}[\mathcal{L}(\Theta_1, \mathcal{B})] + K_{\text{OPT}} \cdot \mathcal{L}(\Theta_{\text{OPT}}, [n] - \mathcal{B})) \cdot (1 + S_{k-1}).$$

Observe that

$$\mathbb{E}_{\Theta_1}[\mathcal{L}(\Theta_1, \mathcal{B})] \leq K_{\text{OPT}} \cdot \mathcal{L}(\Theta_{\text{OPT}}, \mathcal{B})],$$

by Lemma D.2. Moreover, we have that

$$\mathcal{L}(\Theta_{\text{OPT}}, [n] - \mathcal{B}) = \mathcal{L}(\Theta_{\text{OPT}}, [n]) - \mathcal{L}(\Theta_{\text{OPT}}, \mathcal{B}).$$

Using these two expression and noting $S_{k-1} \leq 1 + \log k$, we get the stated result. $\square$

# E  Fair Initialization

We assume the scenario when the user demographic identities are known *a priori*. Let $\gamma : [n] \to \mathcal{A}$ be a mapping, which maps a user $i$ to its demographic group $\gamma(i)$, and let $|\gamma(i)|$ denote the size of the demographic group user $i$ belongs to.

We define the weighted total loss (where each users' loss is divided by its group size) as:

$$\mathcal{G}(\Theta, [n]) = \sum_{i \in [n]} \frac{\mathcal{L}_i(\Theta, \phi_i)}{|\gamma(i)|} = \sum_{i \in [m]} \frac{\mathcal{L}(\Theta, \mathcal{A}_i)}{|\mathcal{A}_i|}.$$

Note that this can also be interpreted as the sum of average losses across different demographic groups.

We introduce Algorithm 2 with a couple of changes from Algorithm 1.

- Instead of uniformly picking a user at random, we are picking a user $i$ with probability proportional to $\frac{1}{|\gamma(i)|}$. This is equivalent to saying, the probability of the first user being in some group $\mathcal{A}_j \in \mathcal{A}$ is

$$\frac{\sum_{i \in \mathcal{A}_j} \frac{1}{|\gamma(i)|}}{\sum_{j \in [m]} \sum_{i \in \mathcal{A}_j} \frac{1}{|\gamma(i)|}} = \frac{1}{m}.$$

  That is, the first user is equally likely to belong to one of the $m$ groups.
- A user $i$ at any time step $t$ is selected with probability proportional to $\mathcal{L}_i(\Theta_{t-1}, \phi_i)/|\gamma(i)|$. This implies that a user from some group $\mathcal{A}_j \in \mathcal{A}$ is likely to be picked with probability proportional to the average demographic loss—i.e.,

$$\sum_{i \in \mathcal{A}_j} \mathcal{L}_i(\Theta_{t-1}, \phi_i)/|\gamma(i)| = \frac{\mathcal{L}(\Theta_{t-1}, \mathcal{A}_j)}{|\mathcal{A}_j|}.$$

---

**Algorithm 2** Fair AcQUIre- Fair Adaptively Querying Users for Initialization

---

1: **Input:** Set of users $[n]$, number of services $k$, Demographic groups $\mathcal{A} = \{\mathcal{A}_1, \ldots, \mathcal{A}_m\}$, Map users to demographic groups $\gamma : [n] \to \mathcal{A}$
2: Choose a user $i$ uniformly randomly from $[n]$ with probability $\propto \frac{1}{|\gamma(i)|}$.
3: Query user $i$'s preference $\phi_i$, set the first service $\Theta_1 = \phi_i$.
4: **for** $t \in \{2, \ldots, k\}$ **do**
5:     **User behavior:** Collect user losses on existing services $\Theta_{t-1} : \{\mathcal{L}_i(\Theta_{t-1}, \phi_i)\}_{i \in [n]}$.
6:     **User Selection:** Sample $l$ from $[n]$ with probability $P(l = i) \propto \mathcal{L}_i(\Theta_{t-1}, \phi_i)/|\gamma(i)|$.
7:     **New service:** Query user $l$'s preference $\phi_l$.
8:     $\Theta_t = \Theta_{t-1} \cup \phi_l$.
9: **end for**
10: **Return** $\Theta_k$

---

We first state the approximation ratio of Algorithm 2 on the objective $\mathcal{G}(\Theta, [n])$.

**Lemma E.1.** *Consider $n$ users with unknown preferences $\{\phi_1, \ldots, \phi_n\} \subset \mathbb{R}^d$, and unknown associated loss functions $\mathcal{L}_i(\cdot, \cdot)$ satisfying Assumptions 2.1 and 2.2. Let $\Theta_{\mathrm{scaled}} \subset \mathbb{R}^d$ be the set of $k$ services minimizing the total loss and $\mathcal{C}(\Theta_{\mathrm{scaled}})$ the resulting partitioning of users. If Algorithm 2 is used to obtain $k$ services $\Theta_k$, then the following bound holds:*

$$\mathbb{E}_{\Theta_k}[\mathcal{G}(\Theta_k, [n])] \leq K_{\mathrm{fair}}(2 + \log k) \cdot \mathcal{G}(\Theta_{\mathrm{scaled}}, [n]),$$

*where the expectation is taken over the randomization of the algorithm and $K_{\mathrm{fair}}$ is equal to*

$$\max_{\mathcal{B} \in \mathcal{C}(\Theta_{\mathrm{scaled}})} \frac{4}{\min\limits_{j \in \mathcal{B}} \sum\limits_{i \in \mathcal{B}} \frac{c_{ij}}{|\gamma(i)|}} \left( \max_{j \in \mathcal{B}} \sum_{i \in \mathcal{B}} \frac{1}{c_{ij}|\gamma(i)|} \right). \tag{10}$$

The proof is similar as Theorem 3.1, with the introduction of $\frac{1}{|\gamma(i)|}$'s when deriving Lemma D.2 and D.3.

**Theorem 3.2.** *Consider $n$ users with unknown preferences $\{\phi_1, \ldots, \phi_n\} \subset \mathbb{R}^d$, and associated loss functions $\mathcal{L}_i(\cdot, \cdot)$ satisfying Assumptions 2.1 and 2.2 with bandit access. Suppose these users belong to $m$ demographic groups $\mathcal{A} = \{\mathcal{A}_1, \ldots, \mathcal{A}_m\} \subset [n]$. Let $\Theta_{\mathrm{fair}} \subset \mathbb{R}^d$ be the set of $k$ services minimizing the fairness objective $\Phi$ given in (4). If Algorithm 2 is used to obtain $k$ services $\Theta_k$, then the following bound holds:*

$$\mathbb{E}_{\Theta_k}[\Phi(\Theta_k, \mathcal{A})] \leq m K_{\mathrm{fair}}(2 + \log k) \cdot \Phi(\Theta_{\mathrm{fair}}, \mathcal{A}),$$

*where the expectation is taken over the randomization of the algorithm and $K_{\mathrm{fair}}$ is defined in* (10).

*Proof.* For any $\Theta \subset \mathbb{R}^d$, we have that

$$\Phi(\Theta, \mathcal{A}) = \max_{i \in [m]} \frac{\mathcal{L}(\Theta, \mathcal{A}_i)}{|\mathcal{A}_i|} \leq \sum_{i \in [m]} \frac{\mathcal{L}(\Theta, \mathcal{A}_i)}{|\mathcal{A}_i|} = \mathcal{G}(\Theta, [n]).$$

For $\Theta_k$—namely the output of Algorithm 1—the above expression implies that

$$\mathbb{E}_{\Theta_k}[\Phi(\Theta_k, \mathcal{A})] \leq \mathbb{E}_{\Theta_k}[\mathcal{G}(\Theta_k, [n])].$$

We also have that

$$\mathcal{G}(\Theta_{\mathrm{fair}}, [n]) = \sum_{i \in [m]} \frac{\mathcal{L}(\Theta_{\mathrm{fair}}, \mathcal{A}_i)}{|\mathcal{A}_i|} \leq \sum_{i \in [m]} \max_{i \in [m]} \frac{\mathcal{L}(\Theta_{\mathrm{fair}}, \mathcal{A}_i)}{|\mathcal{A}_i|} = m \cdot \Phi(\Theta_{\mathrm{fair}}, \mathcal{A}).$$

Let $\Theta_{\mathrm{scaled}}$ be the minimizer of $\mathcal{G}(\cdot, [n])$ and $\Theta_{\mathrm{fair}}$ be the minimizer of $\Phi(\cdot, \mathcal{A})$. Then, we have that

$$\mathcal{G}(\Theta_{\mathrm{scaled}}, [n]) \leq \mathcal{G}(\Theta_{\mathrm{fair}}, [n]) \leq m \cdot \Phi(\Theta_{\mathrm{fair}}, \mathcal{A}) \leq m \cdot \mathbb{E}_{\Theta_k}[\Phi(\Theta_k, \mathcal{A})] \leq m \cdot \mathbb{E}_{\Theta_k}[\mathcal{G}(\Theta_k, [n])].$$

Now using Lemma E.1 for the guarantee on $\Theta_k$ for $\mathcal{G}(\cdot, [n])$, we conclude the proof. $\qquad \square$

*Remark E.2.* If all $c_{ij}$'s are identically equal to some $c > 0$, $K_{\mathrm{fair}} = \frac{4}{c^2}$ and the approximation ratio of Algorithm 2 for the fair objective is $4m(2 + \log k)/c^2$. Meanwhile, Algorithm 1 would have an approximation ratio

$$4m \cdot \frac{\max_{i \in [m]} |\mathcal{A}_i|}{\min_{i \in [m]} |\mathcal{A}_i|} \cdot \frac{(2 + \log k)}{c^2}.$$

# F    Proof of Theorem 4.4

Let the empirical loss—which is a finite sample average regression loss—be denoted

$$\widehat{\mathcal{L}}_i(\theta, \phi_i) = \frac{1}{n_i} \sum_{j \in [n_i]} (\theta^\top x_i^j - y_i^j)^2,$$

where $\{(x_i^j, y_i^j)\}_{j \in [n_i]}$ are **private unknown** features and scores of the users, respectively.

**Lemma F.1.** *Under Assumptions 4.1 and 4.3, each subpopulation empirical loss $\widehat{\mathcal{L}}_i$ satisfies Assumption 2.1 and Assumption 2.2 with*

$$c_{ij} = \frac{1}{2}\lambda_{\min}\left(\frac{\sum_{l \in [n_i]} x_i^l(x_i^l)^\top}{n_i}, \frac{\sum_{l \in [n_j]} x_j^l(x_j^l)^\top}{n_j}\right).$$

*Proof.* Fix a subpopulation index $i$. We start by showing $\widehat{\mathcal{L}}_i$ satisfies Assumption 2.1. We use the following result: if $p \leq d$ random vectors in $\mathbb{R}^d$ are independently drawn from a distribution that is absolutely continuous with respect to the Lebesgue measure, then they are almost surely linearly independent. By Assumption 4.3, applying this result to our scenario, if we draw $n_i \geq d$ features independently from a sub-Gaussian distribution to form a feature matrix $\mathbf{X}_i \in \mathbb{R}^{n_i \times d}$, then it is almost surely full column rank.

Therefore, we compute $\phi_i = \mathbf{X}_i^\dagger y_i$ since $\mathbf{X}_i^\dagger \mathbf{X}_i = \mathbf{I}_p$ when $\mathbf{X}_i$ is full column rank. The subpopulation $i$'s empirical loss is compactly written as $\widehat{\mathcal{L}}_i(\theta, \phi_i) = \|\theta - \phi_i\|_{\mathbf{A}_i}^2$, where $\mathbf{A}_i = \frac{1}{n_i}\sum_{l \in [n_i]} x_i^l(x_i^l)^\top$. Thus $\widehat{\mathcal{L}}_i(\theta, \phi_i)$ satisfies Assumption 2.1.

We now show that $\widehat{\mathcal{L}}_i$ satisfies Assumption 2.2 with $c_{ij} = \frac{1}{2}\lambda_{\min}(\mathbf{A}_i, \mathbf{A}_j)$. To this end, we find the largest $c \in \mathbb{R}_+$ such that $c(u^\top \mathbf{A}_i u) \leq u^\top \mathbf{A}_j u$ and $c(u^\top \mathbf{A}_j u) \leq u^\top \mathbf{A}_i u$ for all $u \in \mathbb{R}^d$.

Rearranging the inequality, this problem is the same as finding the largest $c \in \mathbb{R}_+$ such that

$$u^\top (\mathbf{A}_i - c\mathbf{A}_j)u \geq 0 \quad \text{and} \quad \mathbf{A}_j - c\mathbf{A}_i)u \geq 0 \quad \forall\, u \in \mathbb{R}^d,$$

or equivalently,

$$\mathbf{A}_i - c\mathbf{A}_j \succeq 0 \quad \text{and} \quad \mathbf{A}_j - c\mathbf{A}_i \succeq 0.$$

Therefore, finding such a constant $c$ is easily reformulated as the following optimization problem:

$$\max\{c \mid \mathbf{A}_i - c\mathbf{A}_j \succeq 0, \mathbf{A}_j - c\mathbf{A}_i \succeq 0\} = \lambda_{\min}(\mathbf{A}_i, \mathbf{A}_j).$$

There are two key tools we use to finish the argument.

- For any $u \in \mathbb{R}^d$, $\frac{1}{2}\lambda_{\min}(\mathbf{A}_i, \mathbf{A}_j)\|u\|_{\mathbf{A}_i}^2 \leq \|u\|_{\mathbf{A}_j}^2$.
- For any norm triangle inequality gives us $\|\theta - \phi_i\| \leq \|\theta - \phi_j\| + \|\phi_i - \phi_j\|$. Then we use power mean inequality, i.e. $a \leq b + c \implies a^2 \leq 2(b^2 + c^2)$.

The following shows Assumption 2.2.$(i)$ holds with $c_{ij} = \frac{1}{2}\lambda_{\min}(\mathbf{A}_i, \mathbf{A}_j)$:

$$\frac{1}{2}\lambda_{\min}(\mathbf{A}_i, \mathbf{A}_j)\widehat{\mathcal{L}}_i(\theta, \phi_i) = \frac{1}{2}\lambda_{\min}(\mathbf{A}_i, \mathbf{A}_j)\|\theta - \phi_i\|_{\mathbf{A}_i}^2$$

$$\leq \frac{1}{2}\|\theta - \phi_i\|_{\mathbf{A}_j}^2$$

$$\leq \|\theta - \phi_j\|_{\mathbf{A}_j}^2 + \|\phi_i - \phi_j\|_{\mathbf{A}_j}^2$$

$$= \widehat{\mathcal{L}}_j(\theta, \phi_j) + \widehat{\mathcal{L}}_j(\phi_i, \phi_j).$$

Analogously, the following shows Assumption 2.2.$(ii)$ holds with $c_{ij} = \frac{1}{2}\lambda_{\min}(\mathbf{A}_i, \mathbf{A}_j)$:

$$\frac{1}{2}\lambda_{\min}(\mathbf{A}_i, \mathbf{A}_j)\widehat{\mathcal{L}}_i(\phi_j, \phi_i) = \frac{1}{2}\lambda_{\min}(\mathbf{A}_i, \mathbf{A}_j)\|\phi_j - \phi_i\|_{\mathbf{A}_i}^2$$

$$\leq \lambda_{\min}(\mathbf{A}_i, \mathbf{A}_j)\left(\|\theta - \phi_j\|_{\mathbf{A}_i}^2 + \|\theta - \phi_i\|_{\mathbf{A}_i}^2\right)$$

$$\leq \|\theta - \phi_j\|_{\mathbf{A}_j}^2 + \|\theta - \phi_i\|_{\mathbf{A}_i}^2$$

$$= \mathcal{L}_j(\theta, \phi_j) + \mathcal{L}_i(\theta, \phi_i).$$

This concludes the proof. □

With the preceding technical lemma in place, we now are ready to prove Theorem 4.4.

**Theorem 4.4.** *Suppose users belong to $N$ subpopulations satisfying Assumptions 4.1, 4.2, and 4.3. Let $\{n_i\}_{i \in [N]}$ denote the number of samples per subpopulation. Let $\Theta_k$ be the output of Algorithm 1 using finite sample estimates $\widehat{\mathcal{L}}_i(\cdot, \phi_i)$ and $\Theta_{\mathrm{OPT}}$ be the optimal solution of the expected loss. Then, for any $\epsilon \in (0,1)$, if $n_i = \Omega(\frac{\sigma_i^4 \sqrt{N} \log(2/\delta)}{\epsilon^2})$ for all $i \in [N]$, the following inequality holds with probability at least $1 - \delta$:*

$$\mathbb{E}_{\Theta_k}[\mathcal{L}(\Theta_k, [N])] \le \frac{1+\epsilon}{1-\epsilon} K_{\mathrm{OPT}}(2 + \log k)\mathcal{L}(\Theta_{\mathrm{OPT}}, [N]), \text{ where } K_{\mathrm{OPT}} \text{ is as defined in (3) and}$$

$$c_{ij} = \frac{1}{2} \min \left\{ \lambda_{\min}(\widehat{\Sigma}_i, \widehat{\Sigma}_j), \frac{1}{\lambda_{\min}(\widehat{\Sigma}_i, \widehat{\Sigma}_j)} \right\}, \text{ with } \widehat{\Sigma}_i = \frac{1}{n_i} \sum_{l \in [n_i]} x_i^l (x_i^l)^\top, \ \widehat{\Sigma}_j = \frac{1}{n_j} \sum_{l \in [n_j]} x_j^l (x_j^l)^\top.$$

*Proof.* Lemma F.1 gives us that the subpopulation $i$'s empirical loss for a regressor $\theta \in \mathbb{R}^d$ can be written as $\|\theta - \phi_i\|_{\mathbf{A}_i}^2$. This is a random quantity since $\mathbf{A}_i$ is a sample average covariance of randomly chosen features. We analyse this term next.

For a random feature $x \in \mathbb{R}^d$ in subpopulation $i$, given a fixed center $\theta \in \mathbb{R}^d$, the term $\frac{(\theta - \phi_i)^\top x}{\|\theta - \phi_i\|_{\Sigma_i}}$ is sub-Gaussian with variance proxy $\sigma_i^2$ by Assumption 4.1. The square of a sub-Gaussian random variable, $\frac{(\theta - \phi_i)^\top x x^\top (\theta - \phi_i)}{\|\theta - \phi_i\|_{\Sigma_i}^2}$ is sub-exponential[5] with parameters $\nu_i^2 = 32\sigma_i^4, \alpha_i = 4\sigma_i^2$ (see [22][Appendix B]).

Since subpopulation $i$'s empirical loss is an average of $n_i$ samples, the empirical loss $\|\theta - \phi_i\|_{\mathbf{A}_i}^2$ is a subexponential variable with parameters $\frac{\nu_i^2}{n_i} \|\theta - \phi_i\|_{\Sigma_i}^4, \frac{\alpha_i}{\sqrt{n_i}} \|\theta - \phi_i\|_{\Sigma_i}^2$. This is because the variance of sample average scales as $\frac{1}{\sqrt{n_i}}$, and the constant $\|\theta - \phi_i\|_{\Sigma_i}^2$ scales the subexponential parameters appropriately.

Now, given a set of $k$ arbitrary services $\Theta = \{\theta_1, \ldots, \theta_k\} \subset \mathbb{R}^d$, let $\theta_{\gamma(i)} \in \Theta$ be an arbitrary service chosen by subpopulation $i$. The sum of empirical losses on based on such choices is also subexponential with parameters $\nu^2 = \sum_{i \in [N]} \frac{\nu_i^2}{n_i} \|\theta - \phi_i\|_{\Sigma_i}^4$ and $\alpha = \max_{i \in [N]} \frac{\alpha_i}{\sqrt{n_i}} \|\theta - \phi_i\|_{\Sigma_i}^2$. Using Chernoff bound [40], we have that

$$P\left\{ |\sum_{i \in [N]} \|\theta_{\gamma(i)} - \phi_i\|_{\mathbf{A}_i}^2 - \sum_{i \in [N]} \|\theta_{\gamma(i)} - \phi_i\|_{\Sigma_i}^2| \ge t \right\} \le 2\exp\left(-\frac{t^2}{2\nu^2}\right) \quad \text{where } 0 \le t \le \frac{\nu^2}{\alpha}. \tag{11}$$

Note that the probability bound is is increasing in $\nu$, we thus take an upper bound on $\nu^2$ to upper bound this probability:

$$\nu^2 = \sum_{i \in [N]} \frac{\nu_i^2}{n_i} \|\theta - \phi_i\|_{\Sigma_i}^4 = 32 \sum_{i \in [N]} \frac{\sigma_i^4}{n_i} \|\theta_{\gamma(i)} - \phi_i\|_{\Sigma_i}^4 \le 32 \left(\sum_{i \in [N]} \frac{\sigma_i^8}{n_i^2}\right)^{\frac{1}{2}} \left(\sum_{i \in [N]} \|\theta_{\gamma(i)} - \phi_i\|_{\Sigma_i}^8\right)^{\frac{1}{2}},$$

where the last inequality follows from Cauchy-Schwarz. Further, noticing that $\sum_{i \in n} a_i^4 \le (\sum_{i \in n} a_i^2)^2$, and applying it for the last term twice, we deduce that

$$\nu^2 \le 32 \left(\sum_{i \in [N]} \frac{\sigma_i^8}{n_i^2}\right)^{\frac{1}{2}} \left(\sum_{i \in [n]} \|\theta_{\gamma(i)} - \phi_i\|_{\Sigma_i}^2\right)^2.$$

---

[5]A random variable $x$ is subexponential with parameters $(v^2, \alpha)$ if $\mathbb{E}[\exp(\lambda x)] \le \exp(\frac{\nu^2 \lambda^2}{2}) \ \forall |\lambda| \le \frac{1}{\alpha}$.

Choosing $t = \epsilon \sum_{i \in [N]} \|\theta_{\gamma(i)} - \phi_i\|^2_{\Sigma_i}$ in (11), where $0 \leq \epsilon \leq \mathcal{O}(\min_{i \in [N]} \frac{\sigma_i^2}{\sqrt{n_i}})$, we have that

$$P \left[ | \sum_{i \in [N]} \|\theta_{\gamma(i)} - \phi_i\|^2_{\mathbf{A}_i} - \sum_{i \in [N]} \|\theta_{\gamma(i)} - \phi_i\|^2_{\Sigma_i} | \geq \epsilon \sum_{i \in [N]} \|\theta_{\gamma(i)} - \phi_i\|^2_{\Sigma_i} \right]$$

$$\geq 2 \exp \left( -\frac{\epsilon^2}{64} \left( \sum_{i \in [N]} \frac{\sigma_i^8}{n_i^2} \right)^{-1/2} \right).$$

With high probability, at least $1 - 2\exp \left( -\frac{\epsilon^2}{64} \left( \sum_{i \in [N]} \frac{\sigma_i^8}{n_i^2} \right)^{-1/2} \right)$, for all $\Theta \subset \mathbb{R}^d$, we have that

$$(1 - \epsilon) \sum_{i \in [N]} \|\theta_{\gamma(i)} - \phi_i\|^2_{\Sigma_i} \leq \sum_{i \in [N]} \|\theta_{\gamma(i)} - \phi_i\|^2_{\mathbf{A}_i} \leq (1 + \epsilon) \sum_{i \in [N]} \|\theta_{\gamma(i)} - \phi_i\|^2_{\Sigma_i}.$$

The rest of the proof is deterministic, thus it is assumed everything follows with probability at least $1 - 2\exp \left( -\frac{\epsilon^2}{64} \left( \sum_{i \in [N]} \frac{\sigma_i^8}{n_i^2} \right)^{-1/2} \right)$. Define $\Theta^N = \Theta \times \cdots \times \Theta$—i.e., Cartesian product of the set $\Theta$ $N$-times. Given two functions $f, g : \Theta^N \to \mathbb{R}$, if $f(\theta) \leq g(\theta)$ for all $\theta \in \Theta^N$, then $\min_{\theta \in \Theta^N} f(\theta) \leq \min_{\theta \in \Theta^N} g(\theta)$. Therefore, we deduce that

$$(1 - \epsilon) \sum_{i \in [N]} \min_{j \in [k]} \|\theta_j - \phi_i\|^2_{\Sigma_i} \leq \sum_{i \in [N]} \min_{j \in [k]} \|\theta_j - \phi_i\|^2_{\mathbf{A}_i} \leq (1 + \epsilon) \sum_{i \in [N]} \min_{j \in [k]} \|\theta_j - \phi_i\|^2_{\Sigma_i}. \quad (12)$$

Hence, given the same set of services $\Theta$, the total empirical loss can be bounded with respect to the total expected loss as follows:

$$(1 - \epsilon)\mathcal{L}(\Theta, [N]) \leq \widehat{\mathcal{L}}(\Theta, [N]) \leq (1 + \epsilon)\mathcal{L}(\Theta, [N]).$$

Recall $\Theta_{\mathrm{OPT}}$ denotes the optimal solution for the *total expected loss* and let $\widehat{\Theta}_{\mathrm{OPT}}$ denote the optimal solution for the *total empirical loss*. Let $\Theta_k$ be the output of Algorithm 1 on finite samples per subpopulations. Using the first inequality of (12), we have that

$$(1 - \epsilon)\mathbb{E}_{\Theta_k}[\mathcal{L}(\Theta_k, [N])] \leq \mathbb{E}_{\Theta_k}[\widehat{\mathcal{L}}(\Theta_k, [N])]. \quad (13)$$

Also, noting that $\widehat{\Theta}_{\mathrm{OPT}}$ is the minimizer for $\widehat{\mathcal{L}}(\Theta, [N])$ and using the second inequality of (12), we get that

$$\widehat{\mathcal{L}}(\widehat{\Theta}_{\mathrm{OPT}}, [N]) \leq \widehat{\mathcal{L}}(\Theta_{\mathrm{OPT}}, \Phi) \leq (1 + \epsilon)\mathcal{L}(\Theta_{\mathrm{OPT}}, \Phi). \quad (14)$$

Now combining (13) and (14) with Theorem 3.1, we get the desired result—i.e.,

$$\mathbb{E}_{\Theta_k}[\mathcal{L}(\Theta_k, [N])] \leq \frac{1 + \epsilon}{1 - \epsilon} K_{\mathrm{OPT}}(2 + \log k) \cdot \mathcal{L}(\Theta_{\mathrm{OPT}}, [N]).$$

Setting $\delta = 1 - 2\exp \left( -\frac{\epsilon^2}{64} \left( \sum_{i \in [N]} \frac{\sigma_i^8}{n_i^2} \right)^{-1/2} \right)$, we get that $n_i = \Omega(\frac{\sigma_i^4 \sqrt{N} \log(2/\delta)}{\epsilon^2})$. This concludes the proof.

$\square$

## G Experiment Details

**Census Dataset.** The categorical features (schooling, marital status, migration status, citizenship) were first converted to one hot vectors and the continuous features (age, income) were scaled appropriately for better conditioning. We performed singular value decomposition on the features and retained the top ten components. The scores were taken as the log transform of the daily commute time in minutes. To form subpopulations, we split users based on their Public Use Microdata Area codes (zip code) and ensure each subpopulation belongs to one of the demographic groups considered. The percentage improvements of our algorithm are plotted in Figure 4.

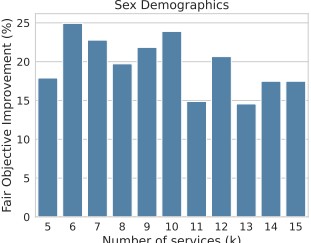
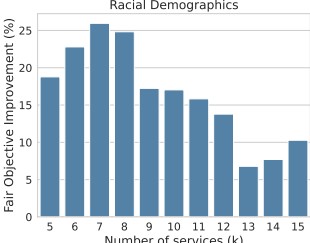

Figure 4: Fair objective improvement for AcQUIre over the baseline across different demographics. We observe that there is atleast 15% improvement across sex demographics for a wide range of number of services. For racial demographics the improvement is 7-26%.

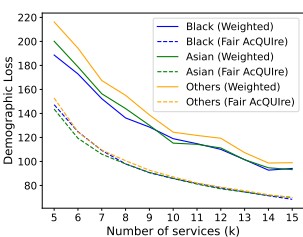
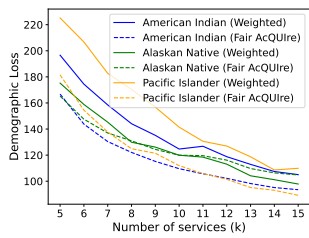
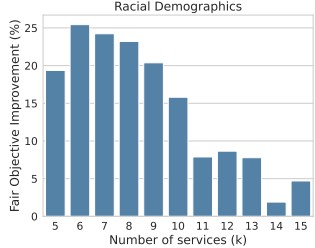

Figure 5: Average losses across different demographic groups for Fair AcQUIre (left,middle). Percentage improvement over baseline (right). We observe that Fair AcQUIre reduces disparity across different groups compared to the baseline.

In Figure 5 we plot the average losses on the individual demographic groups. We benchmark against the weighted random baseline and note that Fair AcQUIre improves not only the fair objective value but the loss for every demographic group.

**Movie Recommendation Dataset.** We use Surprise (a Python toolkit [23]) to perform our experiments. We split the total 10 million ratings into top (a) 200 movies, and (b) all other movies. We use the inbuilt nonnegative matrix factorization function of Surprise on (b) to get user and item embeddings. We cluster the users into 1000 subpopulations by running $k$-means on the obtained user embeddings we get. We evaluate algorithms for this experiment on the held out set (a) of the top 200 movies.

**Ablation:** We use a 2 layer Neural Network with ReLU activations that takes as input the user features and outputs their score. We still use the standard squared prediction error. However note that in this modeling scenario, the loss no longer satisfies our assumptions in the parameters of the neural network. Given a user's features and true score, since there are no unique minimizers, we run gradient descent to compute a local minimizer and then use this trained neural network as a service to predict other user's scores. We run AcQUIre and other baselines under this modeling and report our results in Figure 6. We find that the performance of AcQUIre even when violating assumptions is almost similar to using AcQUIre under modeling which satisfies assumptions.

Additionally, we would like to emphasize that the implementation of our algorithm itself does not rely on these assumptions. It only requires the loss values to be observable. Therefore, one can model very complex precision models and only supply the loss values of these models to our algorithm.

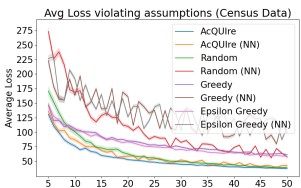 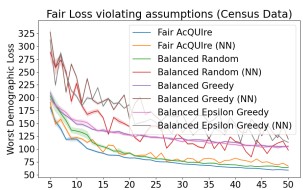 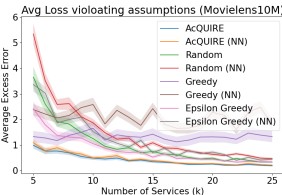

Figure 6: We evaluate the performance of initialization methods when Assumptions 2.1 and 2.2 are violated. "AcQUIre (NN)" refers to using AcQUIre with a Neural Network (NN) to predict scores. This terminology is consistently applied to all other baselines as well. Notably, the performance of AcQUIre doesn't degrade when using Neural Network to predict the scores, thereby demonstrating its robustness to violating the assumptions made in the paper.

