# OpenReview forum: "Initializing Services in Interactive ML Systems for Diverse Users"
_NeurIPS.cc/2024/Conference — NeurIPS 2024 poster_

### Official Review · Reviewer_vEYc · 2024-07-10

**Soundness:** 3
**Presentation:** 3
**Contribution:** 3
**Rating:** 7
**Confidence:** 2

**Summary:**

The paper introduces a novel method for initializing machine learning services tailored to diverse user preferences. The work addresses the challenges of non-convex optimization and lack of pre-existing user preference data before running a service; the authors propose a randomized algorithm that adaptively selects a minimal set of users for data collection. The approach guarantees a total loss close to the global optimum under mild assumptions and extends the k-means++ algorithm to a broader problem class. The results also are supported by experiments on real and semi-synthetic datasets.

**Strengths:**

The authors tackle the challenge of service initialization in the context of bandit feedback and non-convex optimization, which has not been extensively studied before. The proposed algorithm is an interesting extension of the k-means++ algorithm, designed to handle general loss families. The analysis of the proposed approach is solid. The theoretical result is supported by experimental study. The paper is well-written and clearly structured.

**Weaknesses:**

The computational complexity is not well studied, especially in large-scale settings.

**Questions:**

Could you provide any data on the time performance of the algorithms in the experimental study?

---

> ### Author Rebuttal · Authors · 2024-08-07
>
> Thank you for your thorough and thoughtful review of our paper. We appreciate your positive comments on both the theoretical and experimental study, as well as finding our paper well-written. Below, we hope to address your concerns regarding the computational complexity of our algorithm:
>
> We want to emphasize that our algorithm is indeed very computationally efficient, even for large-scale settings like Netflix, where recommendations are made to an order of a billion users. Below, we provide a theoretical analysis of the computational complexity of AcQUIre and also back it up with empirical validation.
> Theoretical Analysis: We sketch the computational complexity of AcQUIre. Each iteration of the loop is examined as follows:
>
> Line 5: Collecting data across a large user base occurs in a distributed network for recommendation systems like Netflix, where
>       queries can be processed in parallel. This step effectively takes O(1) time.
>
> Line 6: After aggregating the data, a new user can be sampled in O(log⁡N) time with modern multinomial samplers (see [1], just as an example).
>
> Lines 7 and 8: These involve O(1) time updates, as they are performed only on the selected user.
>
> Since the loop runs K times, the total cost is O(Klog⁡N). The growth in N is logarithmic, while the growth in K is linear. In practical settings like Netflix recommendation systems, K is much smaller than N. For instance, Netflix groups N=200 million users into 1300 recommendation clusters [2]. Just to compare with our baselines, the greedy and epsilon greedy methods are O(KN) because selecting the highest loss needs going over all N users.
> Experimental Validation: To study the effect of number of users N on the runtime, we generate synthetic datasets in d=200 dimensional space with sizes N=10^4, 10^5, 10^6, 10^7, 10^8, 10^9 (1 billion), and report the runtimes for AcQUIre and the baselines in Figure 2 (left) (attached in the pdf) keeping number of services K fixed at 2000. We observe than even with N=1 billion users, AcQUIre finishes running in less than 300 sec (5 minutes), whereas the greedy and epsilon greedy methods take >10^5 sec (~1day) even for N=10 million users.
>
> To study the effect of the number of services K on the runtime, we generate a dataset with N=5 million users  in d=200 dimensional space  and report the runtimes in Figure 2 (right) as we vary K from 500 to 5000 in steps of 500. We find that even with 5000 services, AcQUIre finishes running in <900 secs (15 minutes), whereas the runtimes for greedy and epsilon greedy are in the range of 10^5 secs (~1-3 days).
>
> We hope this addresses the reviewer’s concerns and demonstrates the practical efficiency of our method for large-scale recommendation systems.
>
> [1] Bringmann, Karl, and Konstantinos Panagiotou. "Efficient sampling methods for discrete distributions." In Automata, Languages, and Programming: 39th International Colloquium, ICALP 2012
>
> [2] https://recoai.net/netflix-recommendation-system-how-it-works/

---

> > ### Comment · Reviewer_vEYc · 2024-08-13
> >
> > Thank you for the addressing my concerns. It sounds satisfactory for me.

---

### Official Review · Reviewer_qH2x · 2024-07-13

**Soundness:** 3
**Presentation:** 3
**Contribution:** 3
**Rating:** 6
**Confidence:** 3

**Summary:**

This paper introduces a novel method for initializing services in interactive machine learning (ML) systems tailored to diverse user preferences. The focus is on scenarios where multiple models or services are deployed, allowing users to choose the one that minimizes their personal losses. The authors highlight two primary challenges in determining optimal initial conditions for these services: the absence of user preference data prior to deployment (bandit feedback) and the presence of non-convex loss landscapes that can lead to suboptimal local solutions. To overcome these challenges, the authors propose a randomized algorithm for service initialization. They provide theoretical guarantees, demonstrating an approximation ratio for the algorithm, and present empirical results that showcase the approach's effectiveness on both real and semi-synthetic datasets.

**Strengths:**

- The proposed adaptive randomized algorithm for service initialization in interactive ML systems is a novel contribution. It extends the well-known K-means++ algorithm to a more complex setting involving diverse user preferences.

- The paper provides strong theoretical guarantees, including tight bounds on total loss and a generalization of the K-means++ guarantee. This adds significant value by ensuring the robustness of the proposed method.

- The empirical results on real and semi-synthetic datasets validate the algorithm's effectiveness in reducing total loss and improving service specialization. The inclusion of fairness considerations further strengthens the practical relevance of the work.

**Weaknesses:**

- The algorithm relies on specific assumptions about the loss functions (e.g., uniqueness of minimizers, approximate triangle inequalities). These assumptions, although reasonable in many cases, may not hold in all practical applications, potentially limiting the algorithm's applicability.

- The empirical validation, although convincing, is limited to two datasets. Additional experiments on a broader range of datasets and application domains would provide stronger evidence of the method's effectiveness and generalizability.

**Questions:**

- What is the performance of optimization algorithms after using your proposed initialization method? Have you empirically compared that performance with other existing initialization approaches?

- Could you elaborate on the potential impact of violating the assumptions made about the loss functions? How robust is your algorithm to deviations from these assumptions in practice?

**Limitations:**

Yes

---

> ### Author Rebuttal · Authors · 2024-08-07
>
> We thank the reviewer for their detailed review, and for finding our tight bounds on a novel setup to be a significant contribution, as well highlighting the practical impacts of our fairness considerations. Below, we hope to address some of the reviewer’s questions:
>
> **Question 1 (Performance of optimization algorithms after initialization):** We thank the reviewer for this question. Once a set of services are initialized, indeed with more user interactions, the provider updates the services on new data to improve the quality (indicated by the reduction in total loss). To evaluate the importance of initialization, we conducted experiments using two different optimization algorithms:
>
> **Generalized k-means:** The services are iteratively updated by training each service on the current group of subpopulations selecting it. After updating the service parameters, the subpopulations reselect their best service. This process repeats until convergence.
>
> **Multiplicative weights update [1]:** Similar to k-means, but each subpopulation can have users choosing different services
>       simultaneously.
>
> Both generalized k-means and the multiplicative weights update guarantee that the total loss reduces over time [1].
>
> In our experiments, we initialized a set of services using our proposed initialization scheme, AcQUIre, and other baseline methods. We then let both optimization algorithms run until convergence. We plotted the total loss values as a function of the number of iterations (see Figure 3 in the attached PDF). Our results demonstrate that our initialization method, AcQUIre, leads to:
>
> **(A) Faster Convergence:** The optimization algorithms converge more quickly with our initialization method compared to other baselines.
>
> **(B) Lower Final Loss:** Initializing with our method convergence to lower losses whereas other initialization schemes are prone to being stuck in suboptimal local minimas. These findings highlight the significance of a robust initialization strategy.
>
> By starting with a better initial configuration, the optimization algorithms can more effectively and efficiently reach higher quality solutions. The reviewer's comments underscore the importance of this aspect, and we believe our empirical comparisons provide strong evidence of the advantages of our proposed method.
>
> **Question 2 (Robustness of AcQUIre to violating assumptions):**
> We appreciate the reviewer’s attention to the potential impact of violating assumptions underlying our proofs. We conduct a new set of experiments, where we use a 2 layer Neural Network with ReLU activations that takes as input the user features and outputs their score. We still use the standard squared prediction error. However note that in this modeling scenario, the loss no longer satisfies our assumptions in the parameters of the neural network. Given a user’s features and true score, since there are no unique minimizers, we run gradient descent to compute a local minimizer and then use this trained neural network as a service to predict other user’s scores. We run AcQUIre and other baselines under this modeling and report our results in Figure 4 (see attached pdf). We find that the performance of AcQUIre even when violating assumptions is almost similar to using AcQUIre under modeling which satisfies assumptions.
>
> Additionally, we would like to emphasize that the implementation of our algorithm itself does not rely on these assumptions. It only requires the loss values to be observable. Therefore, one can model very complex precision models and only supply the loss values of these models to our algorithm.
>
> [1] Dean, Sarah, et al. "Emergent specialization from participation dynamics and multi-learner retraining." International Conference on Artificial Intelligence and Statistics. PMLR, 2024.

---

### Official Review · Reviewer_XL2L · 2024-07-29

**Soundness:** 3
**Presentation:** 3
**Contribution:** 2
**Rating:** 6
**Confidence:** 3

**Summary:**

1) This paper introduces a new algorithm to efficiently initialize a system providing K services to N users (K << N) where user preferences are unknown beforehand and the system iteratively learns about user preferences as the services are recommended, example : netflix movie recommendations.

2) The proposed method has been inspired by the k-means++ algorithm and applies it in a more general setting. Authors also provide theoretical proof that the system initialized in this way will achieve a worst-case loss not exceeding a log multiplier on the optimal loss for this system.

3) AcQUIre and  it's modification to preserve fairness across subpopulations (Fair AcQUIre) are the main algorithms presented in this paper with experiments ablating the effect of different user selection strategies used in AcQUIre

4) Experiments on census dataset and movielens dataset are provided.

**Strengths:**

1) Paper is well-written and authors have done a great job introducing the problem with sufficient notations and related work.

2) The problem introduced is very relevant and encourages future research in this direction.

**Weaknesses:**

1) In the "Movie Recommendation" experiment users are divided into N=1000 subpopulations based on the similarity of their movie ratings and all experiments are then conducted to achieve minimal excess error w.r.t to these subpopulation groups, however in a typical setting where the proposed method might be applied there's no such prior data to group users conveniently so an ablation on user clustering methods prior to applying AcQUIre would further boost it's effectiveness.

2) The size of the datasets are small enough to make the computations practical but the method is actually expensive depending on the choice of K the system needs to get desired loss on the population of N users, so some discussion around this aspect would be useful where authors go deeper in practical deployments of this method is a system like Netflix recommending movies to a billion users.

**Questions:**

1) It is unclear how line 7 of the proposed method : "New service: Query user l’s preference" would be implemented in a real world setting. In a large scale distributed recommendation system such as netflix where user preferences are being collected on a subset of recommended services in parallel as soon as the system is deployed it's unclear how to get a specific user's preferences so some clarification here would be useful.

**Limitations:**

There's no negative societal consequences of this work.

---

> ### Author Rebuttal · Authors · 2024-08-07
>
> We thank the reviewer for their detailed review, and for finding our problem setup novel with the potential to encourage future research in this direction. We appreciate the thoughtful questions the reviewer asked, and believe that our findings and explanations to these questions will further strengthen our paper. Here are our responses:
>
> 1. **Subpopulation grouping (Weakness 1):** We appreciate the reviewer's insight regarding the practical application of our method where prior data for grouping users may not be available. To address this, we have conducted additional experiments to evaluate the dependence of AcQUIre's performance on the quality of grouping. Specifically, we initially form clusters based on prior data, then shuffle x% of the users into incorrect clusters, varying x from 100% (completely random grouping, i.e. no prior data) to 66%, 33%, and 0% (accurate grouping based on prior data). Our results demonstrate  that AcQUIre consistently outperforms the baselines across all values of x (see Figure 1 in attached pdf). Additionally, Figure 1 (right) shows that AcQUIre with different qualities of user grouping outperforms all the baselines with accurate grouping based on prior data. Als0, the performance deterioration as x varies from 0 to 100 is very small, demonstrating AcQUIre’s robustness to accuracy of prior data to form groups.
>
> 2. **Time Complexity (Weakness 2):** We appreciate the reviewer’s concern regarding the computational efficiency of our proposed algorithm, AcQUIre. We want to emphasize that despite these concerns, our algorithm is indeed very computationally efficient, even for large-scale settings like Netflix, where recommendations are made to a billion users. Below, we provide a theoretical analysis of the computational complexity of AcQUIre and also back it up with empirical validation.
> Theoretical Analysis: We sketch the computational complexity of AcQUIre. Each iteration of the loop is examined as follows:
>
>       Line 5: Collecting data across a large user base occurs in a distributed network for recommendation systems like Netflix, where
>       queries can be processed in parallel. This step effectively takes O(1) time.
>
>       Line 6: After aggregating the data, a new user can be sampled in O(log⁡N) time with modern multinomial samplers (see [1], just as an
>       example).
>
>       Lines 7 and 8: These involve O(1) time updates, as they are performed only on the selected user.
>
> Since the loop runs K times, the total cost is O(Klog⁡N). The growth in N is logarithmic, while the growth in K is linear. In practical settings like Netflix recommendation systems, K is much smaller than N. For instance, Netflix groups N=200 million users into 1300 recommendation clusters [2]. Just to compare with our baselines, the greedy and epsilon greedy methods are O(KN) because selecting the highest loss needs going over all N users.
> Experimental Validation: To study the effect of number of users N on the runtime, we generate synthetic datasets in d=200 dimensional space with sizes N=10^4, 10^5, 10^6, 10^7, 10^8, 10^9 (1 billion), and report the runtimes for AcQUIre and the baselines in Figure 2 (left) (attached in the pdf) keeping number of services K fixed at 2000. We observe than even with N=1 billion users, AcQUIre finishes running in less than 300 sec (5 minutes), whereas the greedy and epsilon greedy methods take >10^5 sec (~1day) even for N=10 million users.
>
> To study the effect of the number of services K on the runtime, we generate a dataset with N=5 million users  in d=200 dimensional space  and report the runtimes in Figure 2 (right) as we vary K from 500 to 5000 in steps of 500. We find that even with 5000 services, AcQUIre finishes running in <900 secs (15 minutes), whereas the runtimes for greedy and epsilon greedy are in the range of 10^5 secs (~1-3 days).
>
> We hope this addresses the reviewer’s concerns and demonstrates the practical efficiency of our method for large-scale recommendation systems.
>
> 3. **Querying new user preferences (Question 1):** In a large-scale distributed recommendation system like Netflix, user preferences are collected from initial interactions with the service. When a new set of services is deployed, user preferences can be quickly gathered by prompting users to rate or select their favorite content from a curated list. Additionally, the system can analyze immediate user behaviors, such as search queries, viewing choices, and engagement patterns, to start building a profile. Services like Netflix often also provide incentives such as free first month trial to increase new user engagement during the free trial. As users spend more time watching movies, Netflix gathers more data about their preference.
> It is realistic that all users will not have high levels of engagement when a set of new services are provided. Our algorithm also works when only a small set of a potentially large subpopulation interacts with the provider. Our results in Section 4 provide confidence bounds on the performance of AcQUIre in this setting (see Theorem 4.4).
>
> [1] Bringmann, Karl, and Konstantinos Panagiotou. "Efficient sampling methods for discrete distributions." In Automata, Languages, and Programming: 39th International Colloquium, ICALP 2012
>
> [2] https://recoai.net/netflix-recommendation-system-how-it-works/

---

> > ### Comment · Reviewer_XL2L · 2024-08-13
> >
> > Thanks for addressing each of my questions with detailed experiments. I've gone through the new analysis presented and the attached references. In light of the supporting evidence presented I've raised my score.

---

### Author Rebuttal · Authors · 2024-08-07

We thank all the reviewers for their detailed reviews. We deeply appreciate the questions and insights the reviewer's gave in their reviews. We attach below a set of empirical studies to hopefully answer the questions the reviewers had. Please let us know if we can answer any more questions. Thanks!

---

### Decision · Program_Chairs · 2024-09-25

**Decision:**

Accept (poster)

**Comment:**

The paper gives a new algorithm for an ML system providing K services to N users with service aimed at a sub-group of users. User preferences are unknown beforehand and the system iteratively learns these preferences. The method proposed is inspired by the kmeans++ algorithm. The authors provide a theoretical bound as well as experimental results for their method on cendud dataset and movielens.

The reviewers unanimously appreciated the contribution of the work. Some questions raised by the reviewers were answered sufficiently in the rebuttal. The revision was quite satisfactory to the reviewers. I am happy to recommend acceptance.